# Technical note: On the ice microphysics of isolated thunderstorms and non-thunderstorms in southern China: A radar polarimetric perspective

Chuanhong Zhao[*,1,2], Yijun Zhang[*,1,3], Dong Zheng[4], Haoran Li[4], Sai Du[5], Xueyan Peng[2], Xiantong Liu[5], Pengguo Zhao[2], Jiafeng Zheng[2], Juan Shi[6]

[1]*Department of Atmospheric and Oceanic Sciences & Institute of Atmospheric Sciences, Fudan University, Shanghai, China*

[2]*School of Atmospheric Sciences, Chengdu University of Information Technology, Chengdu, China*

[3]*Shanghai Key Laboratory of Ocean-land-atmosphere Boundary Dynamics and Climate Change & Shanghai Frontiers Science Center of Atmosphere-Ocean Interaction, Fudan University, Shanghai, China*

[4]*State Key Laboratory of Severe Weather, Chinese Academy of Meteorological Sciences & Laboratory of Lightning Physics and Protection Engineering, Chinese Academy of Meteorological Sciences, Beijing, China*

[5]*Guangzhou Institute of Tropical and Marine Meteorology, Guangzhou, China*

[6]*Chengdu Meteorological Office, Chengdu, China*

Corresponding authors: Dr. Yijun Zhang & Dr. Chuanhong Zhao are the co-corresponding authors.
E-mail: zhangyijun@fudan.edu.cn; zch@cuit.edu.cn

**Abstract**

Determining whether a cloud will evolve into a thunderstorm is beneficial for understanding thunderstorm formation and is also important for ensuring the safety of society. However, a clear understanding of the microphysics of clouds in terms of the occurrence of lightning activity has not been attained. Vast field observations and laboratory experiments indicate that graupel, which is rimed ice, is a vital hydrometeor for lightning generation, and is the foundation of riming electrification. In this study, polarimetric radar and lightning observations are used to compare the ice microphysics associated with graupel between 57 isolated thunderstorms and 39 isolated non-thunderstorms, and the differences in radar parameters are quantified. Our results for the occurrence of lightning activity in clouds revealed the following results: 1) the maximum difference in graupel volume at the −10°C isotherm height between thunderstorms and non-thunderstorms reached approximately 7.6 $km^3$; 2) the graupel particles approached spherical shapes with a mean differential reflectivity ($Z_{DR}$) value of 0.3 dB, which likely indicated that heavily rimed graupel was present; 3) the median values of horizontal reflectivity ($Z_H$) or $Z_{DR}$ at positions where the source initiation and channel of the first lightning flashes were nearly 31 dBZ or 0 dB; and 4) 98.2% of the thunderstorms were equipped with a $Z_{DR}$ column, and the mean depth was ~2.5 km. Our study deepens our understanding of lighting physics and thunderstorm formation.

**Short summary**

Understanding lightning activity is important for meteorology and atmospheric chemistry. However, the occurrence of lightning activity in clouds is uncertain. In this study, we quantified the difference between isolated thunderstorms and non-thunderstorms. We showed that lightning activity was more likely to occur with more graupel volume and/or riming. A deeper $Z_{DR}$ column was associated with lightning occurrence. This information can aid in a deeper understanding of lighting physics.

**Keywords**: thunderstorm; lightning; riming; cloud microphysics

## 1. Introduction

Thunderstorms are typically severe convection clouds. Lightning is not only a severe weather hazard produced by thunderstorms but also a clear signature of thunderstorm formation (MacGorman and Rust, 1998). Understanding lightning activity (especially for the first lightning flash, which indicates the start of lightning activity in a cloud) is important for understanding meteorological processes and the formation of thunderstorms (Uman and Krider, 1989; Rosenfeld et al., 2008; Fan et al., 2018) and for investigating related atmospheric chemistry, such as the formation of ozone and the primary oxidant in the troposphere, the hydroxyl radical (Pickering et al., 2016; Brune et al., 2021).

The determination of whether a cloud will evolve into a thunderstorm is very difficult. The occurrence of lightning activity in clouds is a complex process involving dynamics, microphysics and electrical processes (e.g., Krehbiel et al., 1979; MacGorman and Rust, 1998; Carey and Rutledge, 2000; Stolzenburg et al., 2001; Saunders, 2008; Zhang et al., 2009; Lang and Rutledge, 2011; Zhang et al., 2016; Stough and Carey, 2020; Lyu et al., 2023). Moreover, natural lightning flashes are generally defined as intracloud lightning and cloud-to-ground lightning (Uman and Krider, 1989). Some studies have indicated that the majority of the first lightning flashes are intracloud lightning, which was concluded from the statistical results observed by polarimetric radar and lightning location systems (e.g., Mattos et al., 2017; Zhao et al., 2021a). In aadition, there is a generally accepted electrification cause, especially for clarifying the first lightning flash occurrence correctly: noninductive charging (NIC) of two ice particles of different sizes during rebounding collisions in the presence of supercooled droplets, with the smaller ice particle being the ice crystal and the larger ice particle being the graupel; aerosol provides the cloud condensation nuclei and ice nuclei for hydrometeor formation, thus playing an important role in cloud electrification (Takahashi, 1978; Latham, 1981; Saunders et al., 1991; MacGorman and Rust, 1998; Carey and Rutledge, 2000; Rosenfeld et al., 2008; Zhang et al., 2009; Takahashi et al., 2017, 2019; Qie et al., 2021; Lyu et al., 2023).

The NIC was proposed on the basis of cold-chamber laboratory experiments (Reynolds et al., 1957; Takahashi, 1978); subsequently, field observations demonstrated that lightning production is critically linked to ice processes (i.e., graupel signatures) (Dye et al., 1986;

Takahashi et al., 1999; Carey and Rutledge, 2000; Basarab et al., 2015; Stolzenburg et al., 2015; Mattos et al., 2016, 2017; Takahashi et al., 2017, 2019; Hayashi et al., 2021; Zhao et al., 2022). Numerical simulation studies also support the NIC mechanism as the main contributor to charge separation conducive to lightning flash triggering at timescales relevant to storm duration (e.g., Helsdon et al., 2001; Mansell et al., 2005; Barthe and Pinty, 2007). Therefore, graupel is a vital precipitation particle for riming electrification mechanism.

Graupel is rimed precipitation ice. However, the mechanisms for graupel formation vary with cloud type. One pathway to graupel that is very common in warm based clouds worldwide is the development of rain drops in warm rain collision-coalescence processes (e.g., Brahams, 1986; Beard, 1992; Herzegh and Jameson, 1992; Bringi et al., 1997; Carey and Rutledge, 2000), followed by lofting of the rain drop in the updraft to subfreezing temperatures (which is frequently observed by polarimetric radar, called the differential reflectivity ($Z_{DR}$) column), then by drop freezing and finally riming into graupel or small hail. This coalescence-freezing mechanism is often the most important pathway to the first graupel/hail, the first significant electrification and the first lightning flash in warm based clouds (e.g., Brahams, 1986; Beard, 1992; Herzegh and Jameson, 1992; Bringi et al., 1997; Smith et al., 1999; Carey and Rutledge, 2000; Stolzenburg et al., 2015; Mattos et al., 2017). Another pathway to graupel or small hail production is initiated via the aggregation of ice crystals into snow aggregates, followed by riming of the snow aggregate into graupel and possibly even small hail as the rime density increases (Heymsfield, 1982; Li et al., 2018).

It should also be emphasized that the formation of graupel is closely related to not only lightning activity but also the strength of updrafts in clouds, and the latent heat of freezing enhances updrafts, promoting severe storm formation (Rosenfeld, 1999; Zhang et al., 2004; Rosenfeld et al., 2008). More droplets freeze aloft and release more latent heat for nucleation, thereby invigorating convective updrafts and producing lightning, and deep convective clouds form (Rosenfeld, 1999; Zhang et al., 2004; Rosenfeld et al., 2008). Therefore, investigating the ice microphysics associated with graupel is essential for understanding thunderstorm formation.

Polarimetric radar is a better observation system for tracking the specific location and timing of a cloud and inferring the microphysical characteristics within clouds (e.g., Seliga and

Bringi, 1976; Zrnic and Ryzhkov, 1999; Kumjian, 2013; Hu et al., 2019; Huang et al., 2023).
Many studies (e.g., Laksen and Stansbury, 1974; Marshall and Radhakant, 1978; Dye et al., 1986;
Vincent et al., 2003; Latham et al., 2007; Woodard et al., 2012; Mattos et al., 2016, 2017;
Hayashi et al., 2021; Zhao et al., 2022) have investigated the relationship between ice
microphysics and lightning activity and provided methods for predicting the first lightning flash
occurrence based on the riming electrification mechanism; specifically, graupel-related
reflectivity at $-10°C$ or colder is a commonly supported leading reflectivity parameter for
forecasting the first lightning flash (e.g., Laksen and Stansbury, 1974; Marshall and Radhakant,
1978; Vincent et al., 2003; Woodard et al., 2012; Hayashi et al., 2021). However, the
performances of these methods vary with season, geography, or other atmospheric variables;
more directly, different ice microphysics within different clouds dominate. There is no doubt that
the graupel signatures inferred by polarimetric radar are universally present in convective clouds,
whereas some clouds involve no lightning (e.g., Woodard et al., 2012; Hayashi et al., 2021; Cui
et al., 2022; Zhao et al., 2022). Specifically, the graupel signature inferred by the polarimetric
radar needs to be partitioned into more details according to the radar parameters. Therefore, we
conducted this study to better understand the ice microphysics associated with graupel within
thunderstorms.
We accomplished this goal by comparing the ice microphysics associated with graupel
between isolated thunderstorms and non-thunderstorms during the warm season over southern
China and quantifying differences in graupel magnitude and shape (implying the riming
efficiency) in radar parameters, instead of studying the evolution variation within the same
thunderstorm (the role of some polarimetric signatures would be covered in the same cloud
evolution). Furthermore, we discussed the possible microphysics associated with the source
initiation and channel of the first lightning flash via 3D lightning mapping. To our knowledge, no
other study addressing this topic has been published. In addition, we explored the role of the
coalescence-freezing mechanism in the production of lightning based on the information
provided by the $Z_{DR}$ column, a narrow vertical extension of positive $Z_{DR}$ values above the $0°C$
isothermal height associated with updrafts and supercooled liquid water in deep moist convective
storms (e.g., Hall et al., 1980; Ryzhkov et al., 1994; Kumjian and Ryzhkov, 2008; Kumjian, 2013;

141 Kumjian et al., 2014; Snyder et al., 2015; Zhao et al., 2020; Chen et al., 2023). Isolated

142 thunderstorms are common in southern China during the warm season (Mai and Du, 2022). From

143 the perspective of isolated storms in the warm season, the physical processes within clouds are

144 easier to explain, and the characteristics of graupel microphysics can be compared with those of

145 cold-based clouds (Li et al., 2018).

## 2. Materials and methods

147 The dataset used in this study was the same as that used in Zhao et al. (2021a, 2022). In Zhao

148 et al. (2021a), the dataset was first shown to the public, who obtained observations of 57 (39)

149 isolated thunderstorms (non-thunderstorms) that occurred over South China in the warm season

150 (from late May to early September) of 2016 and 2017 from the S-band polarimetric radar and

151 three independent lightning location systems. The role of turbulence characteristics in producing

152 the first lightning flashes was evaluated on the basis of the dataset, and the results indicated that

153 the eddy dissipation rate of non-thunderstorms was clearly lower than that of thunderstorms (Zhao

154 et al., 2021a). Moreover, the polarimetric radar parameters of the first radar echoes (the first radar

155 volume scan when clouds are detected by radar) were compared to determine the early difference

156 between thunderstorms and non-thunderstorms on the basis of this dataset (Zhao et al., 2022). The

157 greater echo intensity occurred in non-thunderstorms below the $-10°C$ isotherm height, and the

158 cause for this feature and effect on subsequent cloud development were simply discussed by

159 integrating comprehensive observations (e.g., the ERA-Interim reanalysis data, surface aerosol

160 concentration, and graupel and rainwater contents derived from radar observations).

161 The error in the graupel content estimated in Zhao et al. (2022) is uncertain, and the

162 efficiency of the microphysical process (i.e., riming) associated with graupel is unknown; this

163 represents a gap in understanding regarding the role of graupel in the first lightning flash

164 occurrence based on field observations. Naturally, we aimed to identify a method to quantify

165 differences in graupel magnitude and riming efficiency in this study to minimize the error as much

166 as possible. The radar sample volume, which corresponds to graupel identification, was used to

167 indicate the graupel magnitude instead of the derived graupel content, as in Carey and Rutledge

168 (2000) and Zhao et al. (2022). The variety of $Z_{DR}$ shapes was used to determine the riming

169 efficiency. Thus, the goal and method of this study were substantially different from those of the

two previous studies noted above, although they are based on the same dataset.
The Guangzhou S-band polarimetric radar (GZ radar) provided the radar data as marked by
the orange star in Figure 1. The beam width of the GZ radar was ≤1°, and a full radar volume scan
lasted 6 minutes; this consisted of nine elevation angles with a radial resolution of 250 m. A
quality control procedure was carried out to remove ground clutter, anomalous propagation, and
biological scatter, and the $Z_{DR}$ offset of the raw data was corrected (Zhao et al., 2022). The
quality-controlled radar data were interpolated onto a Cartesian grid at a horizontal resolution of
250 m and a vertical resolution of 500 m from 0.5 to 20 km above the mean sea level via nearest
neighbour and vertical linear interpolation.

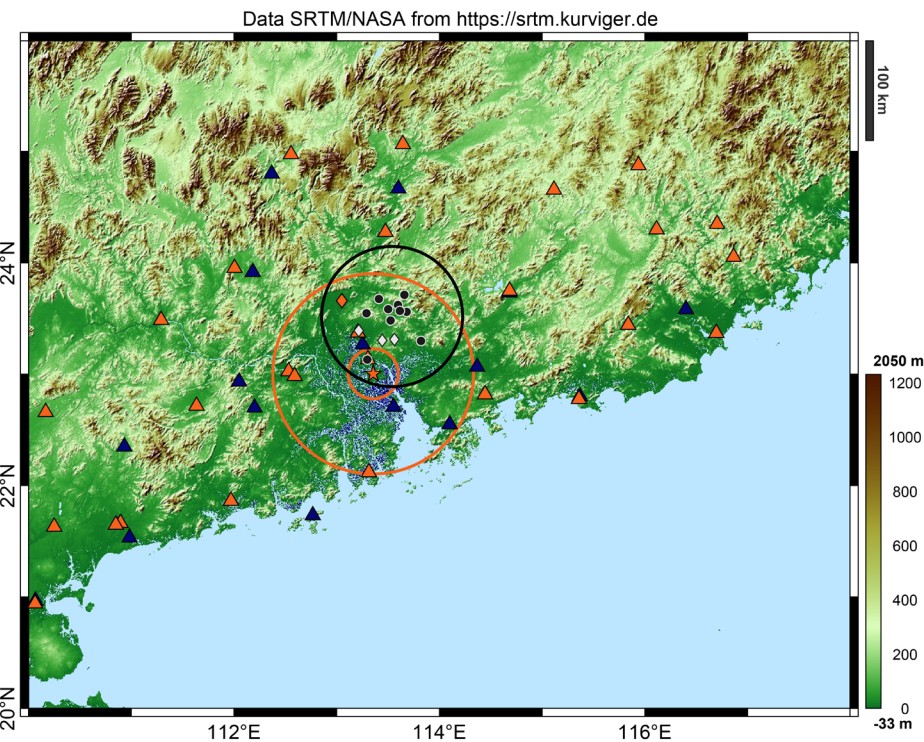


**Figure 1. The locations of the detection systems and the analysed area.** The orange star indicates
the Guangzhou S-band polarimetric radar (GZ radar); the orange circles represent distances from the
GZ radar site of 25 and 100 km. The black dots indicate the 10 sensors of the Low-Frequency E-field
Detection array (LFEDA); the black circle indicates the distance from the centre of the LFEDA
network to 70 km. The blue triangles indicate the 16 sensors of the Earth Networks Lightning Location
System (ENLLS), and the orange triangles indicate the 27 sensors of the Guangdong Lightning
Location System (GDLLS). The white diamonds indicate the three ground sites of aerosol
concentration measurements. The orange diamond indicates the Qingyuan meteorological observatory.
The analysed area is restricted to the regions of overlapping coverage between the GZ radar radius of
25−100 km and the LFEDA station network centre radius of 70 km.

A hydrometeor identification method, which is based on the fuzzy logic algorithm, was carried out to discriminate the graupel particles, as in Zhao et al. (2021b). The algorithm and approximate ranges of the S-band values of each polarimetric variable essentially followed Park et al. (2009) and Kumjian (2013), with an improvement in the parameters of the membership functions of the fuzzy logic algorithm for the performance of the GZ radar, especially for dry/wet snow particles (Wu et al., 2018). In addition, temperature information was one of the few factors added to the hydrometeor identification method because it can separate liquid precipitation from solid hydrometeors to avoid visible identification errors (e.g., Bechini and Chandrasekar, 2015; Kouketsu et al., 2015; Zhao et al., 2020).

Three independent lightning location systems provided lightning observations. The low-frequency E-field detection array (LFEDA, as marked by black dots in Figure 1) can detect three-dimensional structures of intracloud lightning and/or cloud-to-ground lightning. The detection efficiency and mean location error of LFEDA for triggered lightning were approximately 100% and 102 m, respectively (Shi et al., 2017; Fan et al., 2018). The Earth Networks Lightning Location System (ENLLS, as marked by blue triangles in Figure 1) can detect two-dimensional locations for intracloud lightning and/or cloud-to-ground lightning. The detection efficiency and mean location error of ENLLS for triggered lightning and the natural strike of tall structure lightning were approximately 77% and 685 m, respectively (Zheng et al., 2017). The Guangdong Lightning Location System (GDLLS, as marked by orange triangles in Figure 1) can locate cloud-to-ground lightning. The detection efficiency and mean location error of the GDLLS for triggered lightning and the natural strike of tall structure lightning were approximately 94% and 741 m, respectively (Chen et al., 2012).

The lightning flash was assigned to its corresponding cell by using the boundary of the cell as a constraint every 6 minutes. The first lightning flash of a thunderstorm was defined by its first detection from one of three lightning location systems. An isolated non-thunderstorm cell was selected when no flash in the cell was detected by any of the three lightning location systems. To ensure detection-data quality, the analysis area was restricted to the regions of overlapping coverage between the GZ radar radius of 25–100 km and the LFEDA station network centre radius of 70 km (Figure 1), as in Zhao et al. (2021a, 2022). Any isolated cell storm generated within the

analysis area that moved completely outside the analysis area or merged with other precipitation
cells was excluded. The intersection of the 20 dBZ contours of the two intersecting cells is
referred to as merging. For thunderstorms, we ensure that the first lightning flash of the cell must
occur before merging or when there is no merging. For storm cell development, if no merging
process occurs and the maximum reflectivity of this cell starts to fade with a value of less than 30
dBZ later, the evolutionary process of a cell will mark the cessation stage. Our objective was to
focus on isolated storm cells; therefore, if the merging process occurs before the fading of the
maximum reflectivity of this cell, the evolutionary process of the cell will also signal the cessation
stage.
In the dataset, six merging events occurred in non-thunderstorms, and the values of maximum
reflectivity for these non-thunderstorms did not increase after merging occurred. In addition, the
maximum reflectivity within any non-thunderstorm cell from initiation to cessation must exceed
45 dBZ to avoid the statistics of weak precipitation cells. Non-thunderstorms are characterized by
no flash occurrence from initiation to cessation. The sounding data were obtained from the
Qingyuan meteorological observatory, as marked by the orange diamond in Figure 1, which also
provided the environmental temperature. Isolated thunderstorm/non-thunderstorm cells were
identified and tracked manually based on the observations from the GZ radar and lightning
location systems. The average distances between these storms and the radar/sounding site were
approximately 70 and 56 km, respectively. More details related to these data and the selection
methods for isolated thunderstorm and non-thunderstorm cells are available in Zhao et al. (2021a,

239    2022).

In this study, the evolution cycle of a thunderstorm consists of three stages: (i) the first radar
volume scanning in cases where the horizontal reflectivity ($Z_H$) $\geq$ 5 dBZ is called the first stage
(hereafter referred to as the #1 stage), (ii) the intermediate radar volume scanning between the first
stage and the third stage is called the second stage (hereafter referred to as the #2 stage), and (iii)
the radar volume scanning in cases where the first lightning flash occurs is called the third stage
(hereafter referred to as the #3 stage). Similarly, the evolution cycle of a non-thunderstorm also
contains three stages, but radar volume scanning in cases where the most intense echo occurs is
called the third stage; here, the most intense echo is used to indicate the strongest convection
development stage of non-thunderstorms for comparison with the first lightning flash stage of
thunderstorms. The average durations from the first stage to the third stage for thunderstorms and
non-thunderstorms were 19 and 24 minutes, respectively.
The majority of first lightning flash events (~98%) were considered intracloud flashes (IC
flashes), and only one was considered a cloud-to-ground flash (CG flash) (Figure 2a). The
majority of first lightning flashes (~91%) was determined by the LFEDA because of its superior
detection efficiency and accuracy for detecting lightning flashes in this analysis area (Figure 2a).
The elapsed time between the first radar volume scan and the first IC or CG flash (indicated by the
first IC or CG return stroke) is shown in Figure 2b. The results show that the average elapsed time
between the first radar volume scan and the first IC flash was approximately 19 minutes, and the
first CG flash was approximately 32 minutes (Figure 2b). A recent study (Mattos et al., 2017) also
revealed that in ~98% of thunderstorms, an IC flash preceded the first CG flash, and the IC flashes
occurred approximately 29 minutes after the first radar echo (any reflectivity value (any value
above the local noise floor of the radar) at any height), CG flashes were most frequently delayed
by approximately 36 minutes. The definition of the first radar echo may be the possible reason that
the first flashes occurring after the first radar echo in Mattos et al. (2017) occurred later than those
in our study.

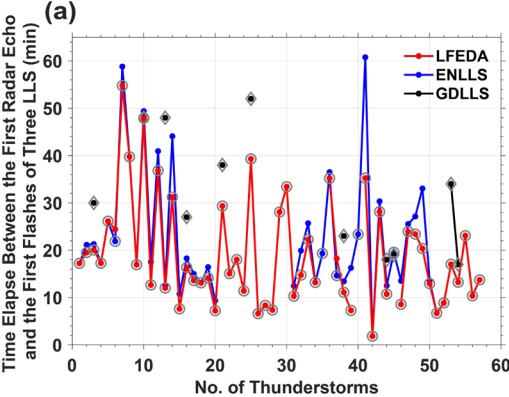 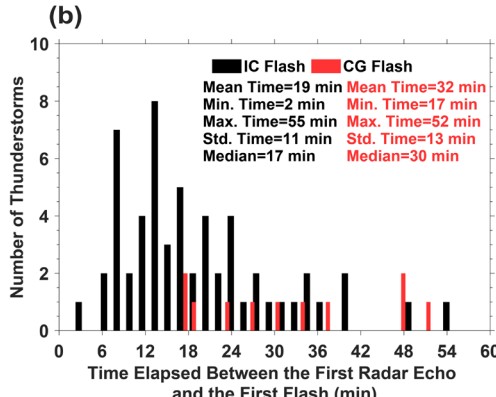

**Figure 2. Lightning observations.** Elapsed time between the first radar volume scan and (a) the first
flashes of three lightning location systems, LFEDA (red line), ENLLS (blue line), and GDLLS (black
line), where the grey circles indicate the first IC flashes, the grey diamonds indicate the first CG flashes,
and (b) the elapsed time between the first radar volume scan and the first flashes of thunderstorms, the
first IC flashes (black columns), and the first CG flashes (red columns).
In addition, the average 1-hourly surface concentration observations of particulate matter
(PM$_{2.5/10}$) were provided by three ground sites (Figure 1, white diamonds) within the analysed area.
The PM$_{2.5/10}$ concentration data suggest that the environment prior to these isolated thunderstorms
or non-thunderstorms was clean and that the difference in the environmental aerosol concentration
between thunderstorms and non-thunderstorms may be small (the mean values of PM$_{2.5/10}$
concentrations prior to thunderstorms and non-thunderstorms were 22.9/42 µg m$^{-3}$ and 20.5/38.8
µg m$^{-3}$, respectively).
**3. Results**
**3.1 Morphology and intensity of the echoes in and/or before the first lightning flash**
**occurrence**
The scatters and triangles with error bars in Figure 3a depict the echo-top heights and
echo-base heights of the 57 thunderstorms and 39 non-thunderstorms from the first stage to the
third stage of cloud development via the reflectivity threshold (0 dBZ), and the echo depths are
shown in the box plots. The echo-top heights of thunderstorms and non-thunderstorms increase as
clouds develop. For the echo-top height data, approximately 95% of the thunderstorms exceeded
the −30°C isotherm height, and 85% exceeded the −38°C isotherm height of the glaciated layer
during the third stage of cloud development; however, only 26% and 23% of the
non-thunderstorms exceeded the −30°C and the −38°C isotherm heights, respectively, during the
third stage of cloud development. However, the echo-base heights mildly decreased with the
development of clouds; slight differences in the echo-base heights occurred between
thunderstorms and non-thunderstorms.
When the first lightning flashes occurred, approximately 84% of the thunderstorms and only
23% of the non-thunderstorms achieved an echo depth of 10 km. Lightning is the product of the
severe storms, and scientists often equate storm intensity with lightning flashes (e.g., Zipser et al.,
2006; Fan et al., 2018), but defining convective intensity is not as easy as it may seem (Zipser et
al., 2006); this could provide supplementary quantitative evidence for assisting scientists in
equating storm intensity with lightning flashes and determining the cloud depth corresponding to
the first lightning flash occurrence.
Figure 3b shows that the differences in the mean (maximum) values of the $Z_H$ between the

thunderstorm and non-thunderstorm periods during each stage are slight; specifically, the median

differences in the mean values are −2, 2, and 3 dBZ, respectively. The median differences in the

maximum values are −4, 5, and 5 dBZ, respectively. Thunderstorms exhibit greater $Z_H$ intensities

than non-thunderstorms do, except for those in the first stage of cloud development. The signature

of larger mean or maximum values of $Z_H$ in non-thunderstorms during the first stage than in

thunderstorms has been discussed by Zhao et al. (2022), and this aspect is not the focus of this

study. The mean or maximum values of $Z_H$ in thunderstorms increase and exceed those in

non-thunderstorms when the first lightning flashes occur; however, the box plots show that we

cannot effectively differentiate thunderstorms from non-thunderstorms with respect to the $Z_H$

intensity.

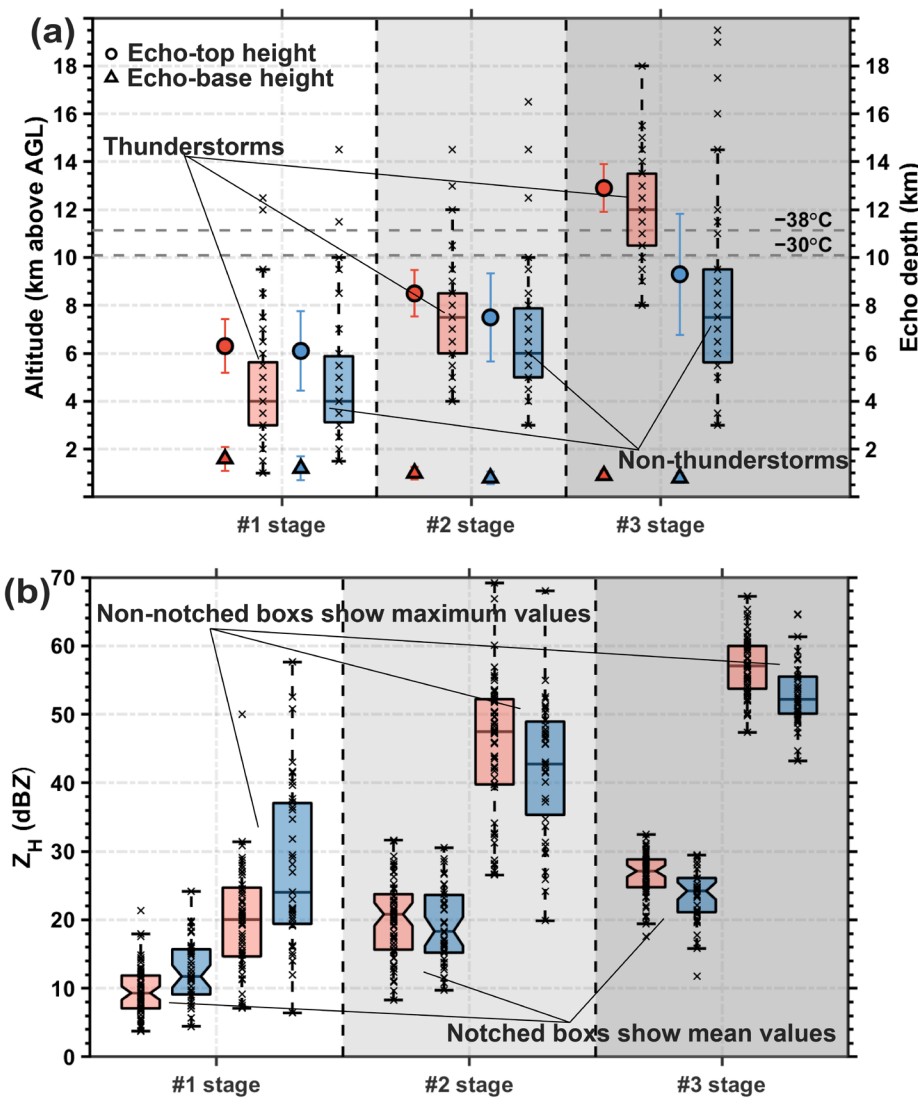

**Figure 3. Characteristics of radar echoes with cloud development.** (a) Echo-top heights of 0 dBZ and echo-base heights of 0 dBZ for 57 thunderstorm and 39 non-thunderstorm cells from the first stage to the third stage of cloud development are indicated by scatter points and triangles, respectively, with error bars. Error bars are computed as 95% confidence intervals. Box plots for the 57 thunderstorms (orange) and 39 non-thunderstorms (blue) for echo depths; all units are in km. The dashed grey lines indicate the −38°C and −30°C isotherm heights. (b) The mean (maximum) value of the $Z_H$ in a thunderstorm or a non-thunderstorm during every stage is shown in notched box plots (non-notched box plots), with all units in the dBZ. The median values in the box plots are shown as black horizontal continuous lines. The temperature data were obtained from the sounding data of the Qingyuan meteorological observatory.

**3.2 Variations in graupel magnitude with cloud development**

Graupel is a vital precipitation particle for the riming electrification mechanism, and its radar signature is not obscured by small ice particles. Thus, to investigate the microphysical characteristics related to the first lightning flash occurrence during storms, we obtained inferred "graupel", which was derived from the fuzzy-logic method based on the GZ radar (Park et al., 2009; Kumjian, 2013; Zhao et al., 2021b, 2022).

Each bar in Figure 4 indicates the mean value of the graupel volume in a height layer (the definition of the height layer is a vertical resolution of 500 m over 0.5 to 20 km above the mean sea level, 40 height layers in total) for 57 thunderstorms or 39 non-thunderstorms during each stage of cloud development. Specifically, the volume is computed by accumulating the radar sample grids; each radar sample grid is 0.03125 $km^3$, 0.25 km×0.25 km×0.5 km.

Graupel is rare in thunderstorms or non-thunderstorms during the first stage of cloud development (e.g., Dye et al., 1986; Mattos et al., 2017), and only 5% (13%) of thunderstorms (non-thunderstorms) show graupel signals (Figure 4). This finding is consistent with the results of Lang and Rutledge (2011), who indicated that the existence of a 30 dBZ echo above the freezing altitude is a necessary condition (in ~90% of cases) for lightning occurrence. This value is well above the 5 dBZ threshold used in this study to detect the first stage of a storm and can explain why graupel is rare in this stage. Moreover, in a modelling study of an isolated thunderstorm, Barthe and Pinty (2007) reported a delay of ~20 minutes between the first occurrence of graupel and the first lightning flash. In this case study, this delay was attributed to the time for graupel and vapour-grown ice to locally gain charge through the NIC mechanism and to the sedimentation of the different particles leading to macroscopic charge separation.

We proposed a mechanism for explaining the larger graupel volume in non-thunderstorms
during the first stage of cloud development: more warm precipitation growth in non-thunderstorms
due to cyclic drop growth resulting from coalescence under weaker updrafts may promote greater
drop formation (Kumjian et al., 2014; Mather et al., 1986; Stough et al., 2021). These larger drops
are lifted above the 0°C isothermal height and freeze to graupel-sized particles via a
coalescence-freezing mechanism (e.g., Bringi et al., 1997; Carey and Rutledge, 2000). With the
development of clouds, that proportion of thunderstorms (non-thunderstorms) that produced
graupel reaches 79% (51%) and 100% (95%) during the second and third stages of cloud
development, respectively.

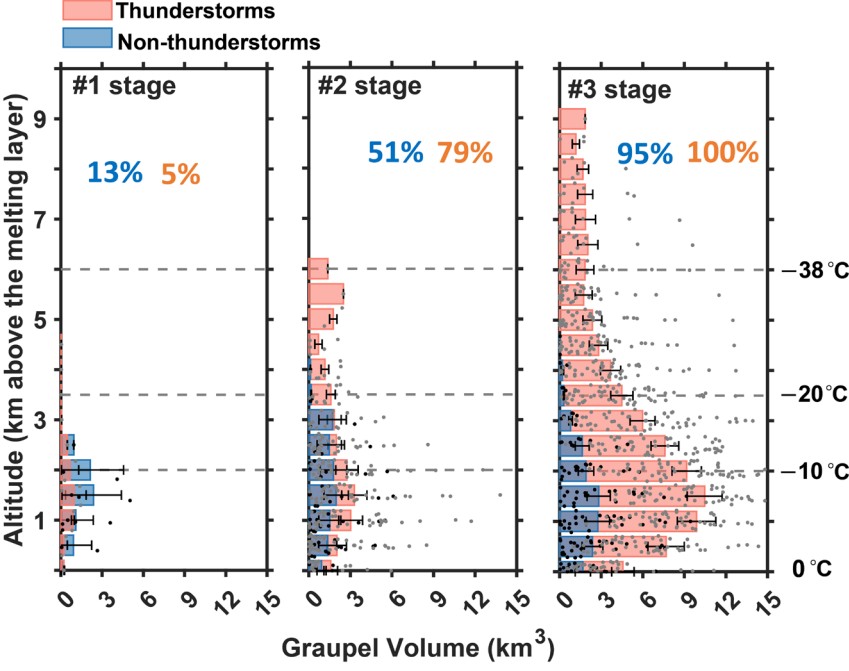


**Figure 4. Distribution of graupel signals and volume with cloud development.** Histogram plots
with error bars for the distribution of the graupel volume above the melting layer for thunderstorm and
non-thunderstorm cells during each stage of cloud development. Each grey dot indicates the total
graupel volume on a height layer (the definition of the height layer is a vertical resolution of 500 m
over 0.5 to 20 km above the mean sea level, 40 height layers in total) of a thunderstorm; the black dots
indicate non-thunderstorms (units in km³). The mean graupel volume in a height layer for the 57
thunderstorms is displayed as an orange histogram and a blue histogram shows the graupel volume for
non-thunderstorm (in km³). Error bars are computed as 95% confidence intervals. The numerical values
in orange and blue are the percentages of thunderstorms and non-thunderstorms that show graupel
signals, respectively. The left column represents the first stage of cloud development, and the right and
middle rows represent the third and second stages of cloud development, respectively. The −10°C,
364                −20°C, and −38°C isotherm heights are displayed in the histogram plots.

The greatest difference in graupel magnitude between thunderstorms and non-thunderstorms
is found during the third stage of cloud development; the maximum difference in graupel volume
in a height layer reaches approximately 7.6 km$^3$, and the height of the maximum difference is near
the −10°C isotherm height. This information is consistent with the NIC electrification mechanism;
namely, more graupel leads to more cloud electrification. In addition, more graupel corresponds to
more latent heat being released for convection invigoration. Interestingly, that the height
corresponding to maximum difference of graupel volume is consistent with the main negative
charge layer in thunderstorms over Guangzhou (Liu et al., 2020). Thus, the results suggested that
the location of the negative charge layer may depend on the height of the maximum graupel
magnitude. Notably, the graupel volume should be more accurately phrased as the presence of
graupel in this volume. These characteristics indicate that graupel signals are universally present in
thunderstorms and non-thunderstorms and that the difference in the magnitude of the graupel
volume is the key for the first lightning flash occurrence.
**3.3 More microphysical information based on radar variables**
As the graupel volume increases from the first radar track to the occurrence of the first
lightning flash, the graupel volume in thunderstorms is clearly greater than that in
non-thunderstorms during the third stage of cloud development. However, the understanding of
the details of the increase in graupel volume is limited (e.g., the variation in the maximum
dimension or number concentration and precursor signature). In addition, although the
coalescence-freezing mechanism dominating the formation of graupel within warm-season
thunderstorms is generally accepted (e.g., Brahams, 1986; Beard, 1992; Herzegh and Jameson,
1992; Bringi et al., 1997; Smith et al., 1999; Carey and Rutledge, 2000; Stolzenburg et al., 2015;
Mattos et al., 2017), more studies are needed to support this mechanism.
The $Z_{DR}$ parameter could provide more information on graupel (e.g., shape) (e.g., Mattos et
al., 2017; Li et al., 2018) and supercooled liquid water (e.g., $Z_{DR}$ column) (e.g., Kumjian, 2013;
Kumjian et al., 2014). The variance in the shape of the graupel indicates the riming efficiency;
specifically, the heavily rimed ice particles approach a spherical shape (Kumjian, 2013; Li et al.,
2018). Although the shape cannot directly indicate the variation in the maximum dimension, the
speculated riming efficiency from the variation in the graupel shape could provide related
information on the maximum dimension of graupel particles; typically, a more spherical shape (a
decrease in $Z_{DR}$) and more riming result in a stronger $Z_H$ corresponding to a larger maximum
dimension (Li et al., 2018). The supercooled liquid water indicated by positive $Z_{DR}$ values above
the 0°C isothermal height is the precursor for freezing particles, followed by the embryo of
graupel particles (e.g., Carey and Rutledge, 2000). Thus, the existence and/or variance of the $Z_{DR}$
column before the occurrence of the first lighting flash could support the coalescence-freezing
mechanism. Moreover, we can obtain the quantitative difference in the $Z_{DR}$ between thunderstorms
and non-thunderstorms, especially for the occurrence of the first lightning flash.
*a. Differences in the shapes of graupel particles between thunderstorms and non-thunderstorms*
The mean values of $Z_H$ and $Z_{DR}$ corresponding to graupel particles (the radar sample grids are
identified as graupel) above the $\sim-3$°C isotherm height (avoiding melting effects) in
thunderstorms and non-thunderstorms during each stage of cloud development are displayed in
Figure 5. Each orange dot indicates the mean values of $Z_H$ and $Z_{DR}$ corresponding to graupel
above the $\sim-3$°C isotherm height in a thunderstorm; each blue dot indicates that in a
non-thunderstorm. On the basis of these results, the average intensity of the $Z_{DR}$ corresponding to
the graupel particles decreases with cloud development, which indicates that the graupel particles
gradually approach a spherical shape (Figure 5d). The most remarkable indicator is that the
graupel particles in the majority of the thunderstorms have lower $Z_{DR}$ values with a mean value of
$\sim0.3$ dB when the first lightning flashes occur; however, this lower $Z_{DR}$ value is not evident in
non-thunderstorms, even during the most intense echo stage of cloud development, with a mean
value of $\sim0.5$ dB. Moreover, the $Z_{DR}$ values approach 0 dB, corresponding to stronger $Z_H$ values
when the average intensity of the $Z_H$ exceeds 35 dBZ. Thus, we speculated that heavily rimed
graupel was present, the size increased, and the shape tended to be spherical.
Li et al. (2018) presented a quantitative relationship between the riming and shape of snow
aggregates in only winter snowstorms; however, we examined the relationship in deep convection
or thunderstorms in the present study. In Li et al. (2018), particles with $Z_H > 15$ dBZ, $Z_{DR} > 0.4$ dB,
and above the $\sim-3$°C isotherm height are likely to be lightly rimed (rime mass fraction $\sim< 0.2$),
and particles with $Z_H > 15$ dBZ, $-0.2 < Z_{DR} < 0.15$ dB, and above the $\sim-3$°C isotherm height are
likely to be moderately or heavily rimed (rime mass fraction $\sim> 0.4$). The rime mass fraction is
defined as the ratio of the accreted ice mass to the total ice particle mass; more details on the rime
mass fraction can be found in Li et al. (2018). In Figures 5a, b, and c, the shaded area in blue
indicates the high possibility that graupel particles are lightly rimed; in contrast, the shaded area in
yellow indicates that the graupel particles are moderately or heavily rimed, as in Li et al. (2018).
The results from Li et al. (2018) are limited to only winter snowstorms; the mechanism for
producing graupel in winter snowstorms is initiated via the aggregation of ice crystals into snow
aggregates, followed by riming of the snow aggregate into graupel and possibly even small hail as
the rime density increases (Heymsfield, 1982; Li et al., 2018). This process is different from the
coalescence-freezing mechanism in warm-season thunderstorms, but the final shape of the graupel
particles when first lightning flashes occurred in this study approached the shape of moderately or
heavily rimed ice particles in Li et al. (2018).

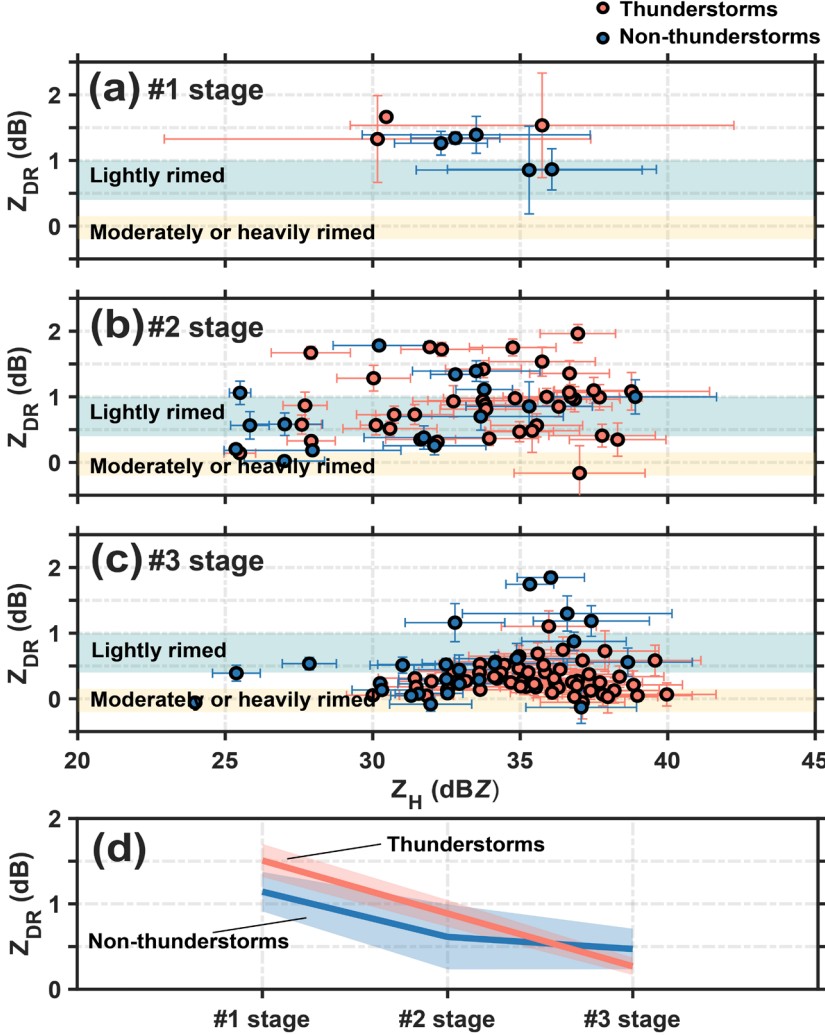


**Figure 5**. **Graupel shape in and/or before the first lightning flash occurrence.** Scatter plots with
error bars for the mean values of $Z_H$ and $Z_{DR}$ corresponding to graupel particles above the $\sim-3°C$
isotherm height in thunderstorm (orange) and non-thunderstorm (blue) cells during each stage of cloud
development. Error bars are computed as 95% confidence intervals. The inferred differences in the
efficiency of the riming process are shown by the threshold values of $Z_H$ and $Z_{DR}$; the shaded area in
blue indicates the high possibility that graupel particles are lightly rimed, and comparatively, the
shaded area in yellow indicates that graupel particles are moderately or heavily rimed. (a) First stage, (b)
second stage, and (c) third stage of cloud development. In addition, the statistical mean values are
given in (d), and the orange (blue) line indicates the mean value of the $Z_{DR}$ corresponding to the above
scatters in thunderstorms (non-thunderstorms) during each stage of cloud development. The shaded
area indicates the 95% confidence interval.

b.  *Observational characteristics associated with the source initiation and channel of the first*
*lightning flash*

The characteristics at positions with source initiation and channel characteristics of the first

lightning flash are shown in Figure 6, including the height distribution, associated hydrometeor
type, and values of $Z_H$ and $Z_{DR}$. The heights of the initiation sources and propagation sources of
the first lightning flashes determined via LFEDA are concentrated at an approximate −10°C
isotherm height (Figure 6a), which is consistent with the results (i.e., the negative charge layer is
located at 6 to 8 km height in thunderstorms over Guangzhou) reported by Liu et al. (2020). The
hydrometeor types associated with the initiation and propagation sources are similar, and the
majority of these particles are graupel and ice crystals (Figure 6b), which is understandable on the
basis of the NIC electrification mechanism.

The median values of $Z_H$ are near 31 dBZ, and the $Z_{DR}$ values are near 0 dB (Figure 6c, d).

Furthermore, Figure 7 displays the frequency of initiation and propagation sources corresponding
to value intervals of $Z_H$ (4 dBZ) and $Z_{DR}$ (0.2 dB). The results indicate that the initiation sources of
the first lightning flashes likely correspond to 20~40 dBZ and −0.2~0.4 dB (Figure 7a), and the
values are likely 16~44 dBZ and −0.2~0.8 dB from propagation sources, respectively (Figure 7b).

These characteristics provide supplementary evidence that the main negative charge layer is

located at −10°C to −20°C isotherm height on Earth, as reported by Krehbiel (1986), and suggest
that are differences in particle shape and/or size between initiation sources and propagation
sources, although the differences are too subtle to quantify in this study.

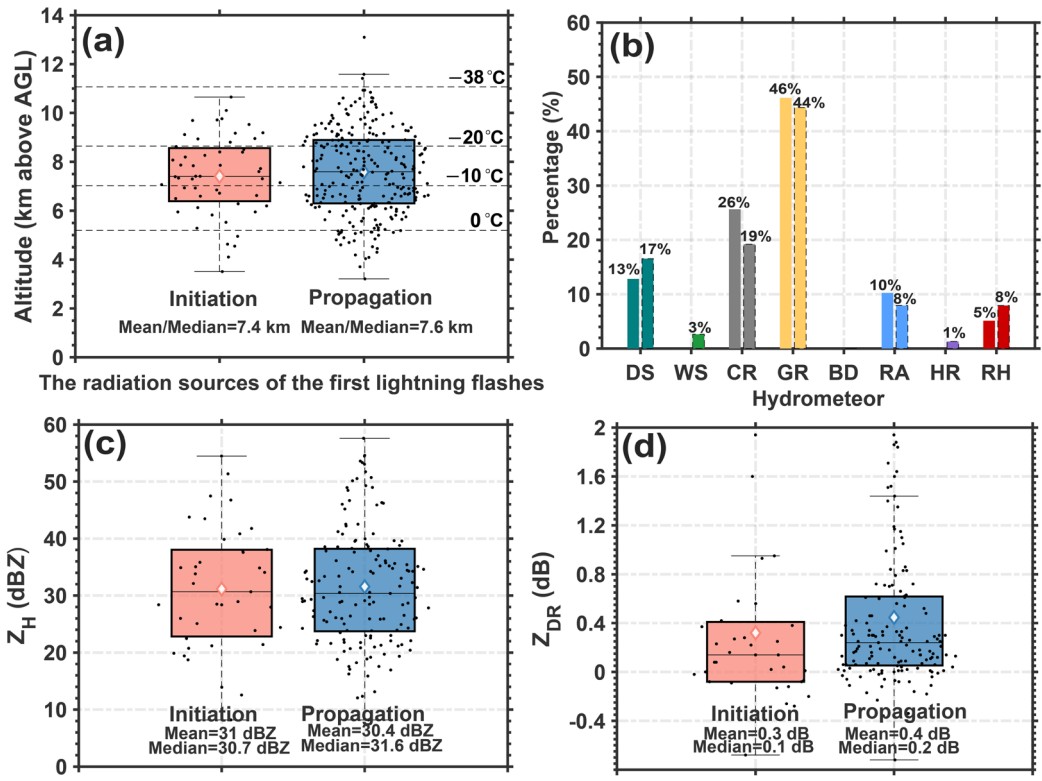

**Figure 6. The characteristics at positions with source initiation and the channel of the first lightning flash.** (a) Height distribution of the locations at the initial sources (orange box) or propagation sources (blue box) of the first lightning flashes. The 0°C, −10°C, −20°C, and −38°C isotherm heights are displayed. (b) The histogram indicates the percentage of various hydrometeors of the locations at the initial sources or propagation sources (histogram with dashed line) of the first lightning flashes. The numerical value is the percentage of various hydrometeors, such as dry snow (DS, dark green), wet snow (WS, green), crystals (CR, grey), graupel (GR, yellow), big drops (BD), raindrops (RA, blue), heavy rain (HR, purple), and rain and hail mixtures (RH, red). Radar parameters of the locations at the initial sources (orange box) or propagation sources (blue box) of the first lightning flashes: (c) horizontal reflectivity ($Z_H$) and (d) differential reflectivity ($Z_{DR}$). Each black dot indicates an individual source. The diamonds indicate the mean values.

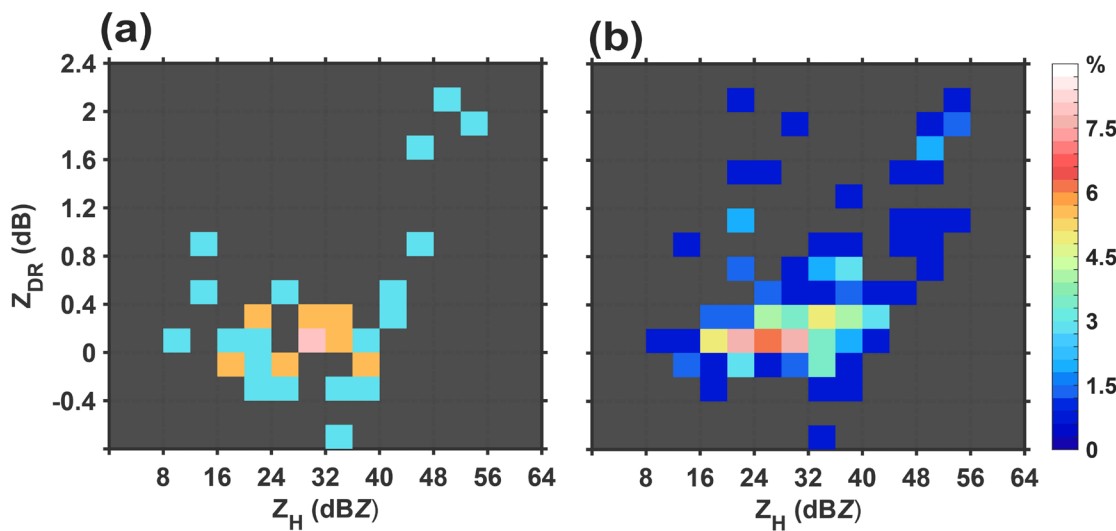

**Figure 7**. **The frequency of radiation sources corresponding to the value intervals of $Z_H$ and $Z_{DR}$.**
(a) Initial sources. (b)Propagation sources.

*c.   Signature of the $Z_{DR}$ column*

Previous studies utilized $Z_{DR}$ values ranging from 0.5–5 dB within the strong reflectivity range (35–50 dBZ) above the melting layer to describe the area of the $Z_{DR}$ column (e.g., Illingworth et al., 1987; Tuttle et al., 1989; Ryzhkov et al., 1994; Scharfenberg et al., 2005; Woodard et al., 2012; Kumjian et al., 2014; Snyder et al., 2015; Zhao et al., 2020). Since the development of these clouds in this study occurred during the early stage of the full evolution cycle of thunderstorms, the size of the supercooled liquid water drop would not be large. Thus, we used $Z_{DR}$ values of 0.5 dB within a reflectivity range of 30 dBZ above the melting layer to investigate the characteristics of the $Z_{DR}$ column.

Figure 8 shows the height of the $Z_{DR}$ column within thunderstorms or non-thunderstorms during each stage of cloud development. The computation of the $Z_{DR}$ column height is similar to that in Snyder et al. (2015), and this height is the vertically continuous maximum depth of the $Z_{DR}$ column. The signature of the $Z_{DR}$ column clearly coincides with the development of clouds (Figure 8). Most thunderstorms (98.2%) displayed a deep $Z_{DR}$ column with a mean depth of the $Z_{DR}$ column of ~2.5 km when the first lightning flash occurred; however, only 48.7% of non-thunderstorms corresponded to a shallow $Z_{DR}$ column with a mean value of ~1.1 km (Figure 8a, b). Moreover, 66.7% of the thunderstorms presented a deeper $Z_{DR}$ column with a mean value of ~1.5 km during the second stage of cloud development, and 30.8% of the non-thunderstorms presented a shallower $Z_{DR}$ column with a mean value of ~0.99 km during the second stage of cloud development (Figure 8a, b).

The results indicate that strong relationship between the $Z_{DR}$ column and the occurrence of the first lightning flash is persistent. A deeper $Z_{DR}$ column suggests a greater graupel volume. However, the occurrence frequency of the $Z_{DR}$ column for non-thunderstorms is slightly greater than that for thunderstorms during the first stage of cloud development (Figure 8a, b). This phenomenon may be related to the results of Zhao et al. (2022); specifically, the $Z_{DR}$ values below the −10°C isotherm height of non-thunderstorms were greater than those of thunderstorms within the first radar echo.

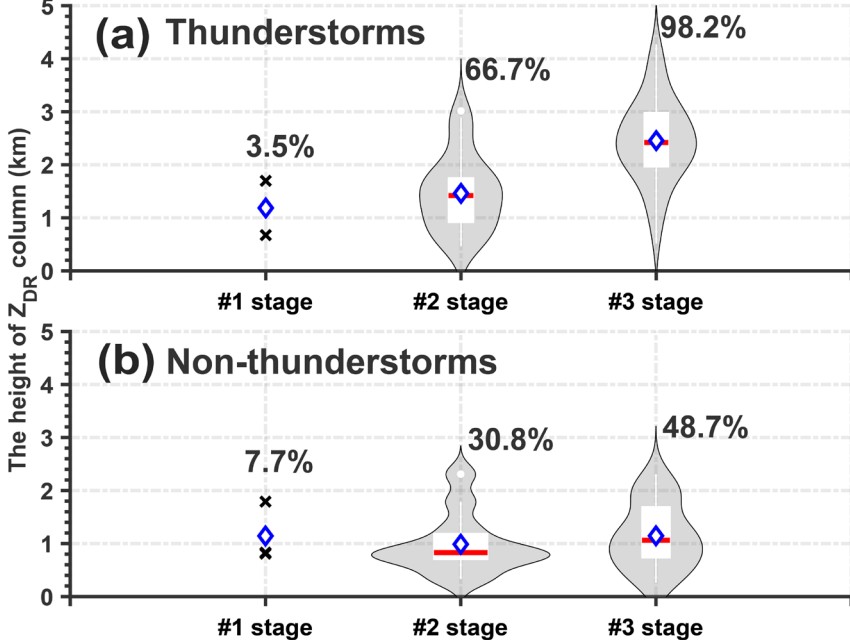

**Figure 8**. **$Z_{DR}$ column information in and/or before the first lightning flash occurrence.** Violin plots of the $Z_{DR}$ column depth of thunderstorm or non-thunderstorm cells during each stage of cloud development, showing the average (blue diamond), interquartile range (rectangle), 10th and 90th percentiles (whiskers), and kernel density estimation (gray shading). (a) Thunderstorms. (b) Non-thunderstorms. The numerical value is the percentage of thunderstorms that show the $Z_{DR}$ column signature.

**4. Summary**

In this study, a combination of a lightning location system and dual-polarization radar measurements was employed to study the ice microphysics of isolated thunderstorms and non-thunderstorms in southern China during the warm season. From the unique perspective of comparing radar signatures and inferred graupel information between isolated thunderstorm and non-thunderstorm cells during each stage of cloud development, lightning generation in clouds was found to be a good indicator of the formation of deep convective clouds. The echo intensities, echo-top heights and echo depths were greater in clouds when the first lightning flash occurred, which indicated more severe updrafts in thunderstorms than in non-thunderstorms. Moreover, a greater graupel volume was clearly observed in clouds when the first lightning flash occurred, and the maximum difference in graupel volume in the height layer between thunderstorms and non-thunderstorms reached approximately 7.6 km³, corresponding to an approximate −10°C isotherm height.

529   The variation in the average $Z_{DR}$ intensity corresponding to the graupel particles above the

530  $\sim-3°C$ isotherm height during the three stages of cloud development indicated that graupel

531  particles were more spherical (the mean $Z_{DR}$ value was ~0.3 dB) and were more likely to generate

532  lightning. The $Z_{DR}$ values approached 0 dB, corresponding to stronger $Z_H$ values; the average

533  intensity of the $Z_H$ exceeded 35 dBZ. When the first lightning flashes occurred in clouds, a

534  decrease in the $Z_{DR}$ value and an increase in the $Z_H$ value of graupel were observed; these results

535  indicate that heavily rimed ice particles were present and that the shape of these particles was

536  similar to that of moderately or heavily rimed ice particles within winter snowstorms.

537   Furthermore, observational characteristics associated with the source initiation and channel of

538  the first lightning flash were investigated. The results revealed that these sources were

539  concentrated at an isotherm height of approximately $-10°C$ and mainly corresponded to graupel

540  and ice crystals. The median values of $Z_H$ or $Z_{DR}$ at the positions of source initiation and the

541  channel of the first lightning flashes were nearly 31 dBZ or 0 dB. In addition, we suggest that the

542  differences in particle shape and/or size between the initiation sources and propagation sources of

543  the first lightning flashes persist.

544   Moreover, the results indicated a strong relationship between the $Z_{DR}$ column and the

545  occurrence of the first lightning flash; 98.2% of the clouds were equipped with a $Z_{DR}$ column with

546  a mean depth of ~2.5 km when the first lightning flash occurred. In addition, a deeper $Z_{DR}$ column

547  corresponded to a greater graupel volume. Thus, the coalescence-freezing mechanism dominated

548  the formation of graupel within warm-season isolated thunderstorms over southern China, and the

549  results were consistent with those of previous studies (e.g., Brahams, 1986; Beard, 1992; Herzegh

550  and Jameson, 1992; Bringi et al., 1997; Smith et al., 1999; Carey and Rutledge, 2000; Stolzenburg

551  et al., 2015; Mattos et al., 2017) but increased the knowledge of the quantified characteristics of

552  the $Z_{DR}$ column for the first lightning flash occurrence in warm-season isolated thunderstorms on

553  the basis of relatively large sample statistics (Table 1 shows details of cases in related

554  investigations for isolated thunderstorms).

| References | Number of cases (thunderstorms) | Number of cases (non-thunderstorms) |
|---|---|---|
| Workman and Reynolds, 1949 | 12 | × |

| | | |
|---|:---:|:---:|
| Reynolds and Brook, 1956 | 5 | × |
| Goodman et al., 1988 | 1 | × |
| Ramachandran et al., 1996 | 2 | × |
| Jameson et al., 1996 | 3 | × |
| Woodard et al., 2012 | 31 | 19 |
| Stolzenburg et al., 2015 | 3 | × |
| Mattos et al., 2017 | 46 | × |

Table 1. Details of the cases in the references.

However, our results were obtained by comparing the characteristics of the polarimetric

parameters according to the graupel particles inferred via a hydrometeor identification method.
The inferred graupel volume was an indication that graupel could be present among other
hydrometeors in that volume. From the perspective of radar, the dominant particle in this volume
was graupel. Fortunately, we focused on comparing the graupel volume between thunderstorms
and non-thunderstorms; therefore, we believe that the errors in this volume resulting from other
secondary hydrometeors could be neutralized by comparisons with the same detected data and
methods.

In addition, unlike previous similar studies (e.g., Mattos et al., 2016, 2017), we studied the

microphysical differences between isolated thunderstorms and non-thunderstorms during the
warm season over southern China on the basis of polarimetric radar and lightning mapping array
instead of studying the evolution variation within the same thunderstorm (Mattos et al., 2017) or
studying the differences between storm vertical profiles in three-dimensional Cartesian boxes with
lightning and without lightning (Mattos et al., 2016).

Although the results from this study could provide a possible index or method based on

polarimetric radar for warning of the first lightning flash occurrence within warm-season cell
storms, understanding the microphysical characteristics and applying that in the numerical
simulations would be the optimal method for providing lightning flash warnings in the future.



**Acknowledgements**

The authors acknowledge the Guangzhou Institute of Tropical and Marine Meteorology for collecting and archiving the radar, the surface, and the lightning observations. And authors also acknowledge the State Key Laboratory of Severe Weather, Chinese Academy of Meteorological Sciences & Laboratory of Lightning Physics and Protection Engineering for three-dimensional lightning location data. This research has been supported by the National Natural Science Foundation of China (grants 42175090, 42305079, 42305087), the China Postdoctoral Science Foundation (grant 2023M730619), the Scientific Research Fund of Chengdu University of Information Technology (grants KYTZ202213, KYQN202301, KYQN202307), the Scientific Research Fund of CAMS State Key Laboratory of Severe Weather (2021LASW-B02), and Basic Research Fund of CAMS (451490, 2023Z008).

**Open Research**

The sounding data is available at http://weather.uwyo.edu/upperair/sounding.html. The data in this study can be obtained from Figshare (Zhao, 2024).

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

Funding acquisition: Y. Zhang, C. Zhao
Investigation: C. Zhao, Y. Zhang
Methodology: C. Zhao, Y. Zhang, and H. Li
Project Administration: Y. Zhang
Resources: C. Zhao, Y. Zhang
Software: C. Zhao, D. Zheng
Supervision: Y. Zhang
Validation: C. Zhao, Y. Zhang
Visualization: C. Zhao, Y. Zhang, and H. Li
Writing-original draft: C. Zhao, Y. Zhang, X. Peng, and H. Li
**Competing interests**
The contact author has declared that none of the authors has any competing interests.