# Peer review of "Technical note: On the ice microphysics of isolated thunderstorms and non-thunderstorms in southern China: A radar polarimetric perspective"

_EGUsphere, 2024_

## Author Comment (AC1)

**Responses to Reviewers' Comments**

We sincerely appreciate the time and effort devoted by the anonymous reviewers and editor. We thank the reviewers for these constructive and professional comments. And Our point-to-point responses can be found below. The reviewer comments/suggestions are in *italic* font, and our responses are underlined and in blue. The file name "Manuscript with marked changes" is abbreviated as "mms".

**Referee #1 Evaluations:**

*This technical note provides valuable information on the characteristics of thunderstorms and non-thunderstorms cells based on polarimetric radar and lightning observations over South China.*

*The manuscript is well written, the methodology is clear, and the results are correct and valuable. They provide quantitative data on the conditions that lead to the occurrence of lightning, and can be helpful to improve lightning forecasting systems. I have some minor concerns before the technical note can be published:*

*- General: The thunderstorms and non-thunderstorms cell analysed in this study have already been analysed by the authors in two previously published studies, as acknowledged here. In particular, the ZH, the ZDR, the content of graupel and the graupel shape has already been analysed in Zhao et al. (2022, GRL). In this study, the ice microphysics associated with graupel is studied by comparing different thunderstorms and non-thunderstorm cells instead of by only studying the evolution of particular cells. This is novel. However, I would appreciate a more detailed description of the novelty of this work with respect to the two previous studies.*

*In addition, I would appreciate more information about the analysis of these thunderstorms from the two previously published papers. For example, while reading this technical note I wondered if the content of aerosols could pay a role in this analysis, as they are not mentioned here. Later, I noted that the content of aerosols in these thunderstorms was analysed in Zhao et al. (2022, GRL). Mentioning this in this paper could help the reader understanding the analysed thunderstorms.*

**Reply:** We sincerely appreciate for your evaluation and insightful comment. We have added more description of the novelty of this work with respect to the two previous studies (Zhao et al., 2021a, 2022). Moreover, more information about the analysis of these thunderstorms from the two previously published papers is provided in the revised manuscript as suggested. Please see in mms (Lines 137−138; 150−175).

The information about the content of aerosols has been added to the revised manuscript. Please see mms (Lines 278−284).

This study is substantially different from the two previous studies (i.e., Zhao et al., 2021a, 2022) noted above although they are used the same dataset.

In Zhao et al. (2021a), we first presented the dataset to the public, which included observations of 57 (39) isolated thunderstorms (non-thunderstorms) during 2016/2017 over South China from the S-band polarimetric radar and three independent lightning location systems. The objective of this study was to investigate the turbulence characteristics of thunderstorms before the first flash in comparison to those of non-thunderstorms. We utilized this dataset and derived the eddy dissipation rate from the Doppler velocity to evaluate the role of turbulence characteristics in producing the first lightning flash in the cloud. The results indicated that the eddy dissipation rate of non-thunderstorms was clearly lower than that of thunderstorms.

At the peer review stage of Zhao et al. (2021a), an anonymous reviewer noted the turbulence difference in the first radar volume scan between thunderstorms and non-thunderstorms (i.e., a stronger eddy dissipation rate in non-thunderstorms). However, we also propose the following question: what was the difference between thunderstorms and non-thunderstorms in the first radar volume scan, and did it affect cloud development?

Thus, we utilized this dataset to evaluate the polarimetric radar parameters of the first radar echoes (the first radar volume scan when clouds occurred and were detected by radar) in Zhao et al. (2022). We discovered that the polarimetric radar parameters of the first radar echoes clearly differed between thunderstorms and non-thunderstorms; specifically, a greater echo intensity was present in non-thunderstorms below the $-$10°C isotherm height. In addition, the ERA-Interim reanalysis data and surface aerosol concentration observations were used to determine the reason. Finally, the graupel and rainwater contents (the value of the 90% quantile at different altitudes during different development stages of storms) were compared, and the results suggested that the difference in the first radar echoes between thunderstorms and non-thunderstorms may play an important role in subsequent cloud development.

In Zhao et al. (2022), the difference in polarimetric radar parameters in the first radar echoes between thunderstorms and non-thunderstorms was determined. In addition, the graupel content was shown during cloud development to suggest convection invigoration according to latent heat release.

However, the error in graupel content estimation is uncertain, and the efficiency of the microphysical process (i.e., riming) associated with graupel is unknown. Naturally, we want to seek a method to quantify differences in graupel magnitude and riming efficiency, while minimizing the error as much as possible.

Therefore, we accomplish this goal by comparing the ice microphysics associated with graupel between isolated thunderstorms and non-thunderstorms during the warm season over southern China and quantifying differences in graupel magnitude and shape (implying riming efficiency) in radar parameters. The radar sample volume, which corresponds to graupel identification, is used to indicate the graupel magnitude instead of the derived graupel content, as in Carey and Rutledge (2000) and Zhao et al. (2022). The variety of $Z_{DR}$ shapes is used to determine the riming efficiency. In addition, the coalescence-freezing mechanism, which is a generally accepted mechanism for graupel formation in warm-based clouds, is explored for the production of the first lightning flash. The results (i.e., the variety of $Z_{DR}$ shapes) could be compared with those in cold-based clouds (Li et al., 2018).

Moreover, in the revised manuscript, the observational characteristics of the first lightning flashes are shown via 3D lightning mapping from LFEDA. The possible microphysics associated with the source initiation and channel of the first lightning flashes are discussed.

It should be noted that the graupel shape is first time analysed in this study, we quantified differences of graupel shape (the change of $Z_{DR}$, implying the riming efficiency) between isolated thunderstorms and non-thunderstorms during the different stages of cloud development.

Therefore, the goal and method in this manuscript is substantially different from the two papers noted above, although they are based on the same dataset.

**Lines 137−138** in mms:

"Furthermore, we discussed the possible microphysics associated with the source initiation and channel of the first lightning flash via 3D lightning mapping."

**Lines 150−175** in mms:

"The dataset used in this study was the same as that used in Zhao et al. (2021a, 2022). In Zhao et al. (2021a), the dataset was first shown to the public, who obtained observations of 57 (39) isolated thunderstorms (non-thunderstorms) that occurred over South China in the warm season (from late May to early September) of 2016 and 2017

from the S-band polarimetric radar and three independent lightning location systems. The role of turbulence characteristics in producing the first lightning flashes was evaluated on the basis of the dataset, and the results indicated that the eddy dissipation rate of non-thunderstorms was clearly lower than that of thunderstorms (Zhao et al., 2021a). Moreover, the polarimetric radar parameters of the first radar echoes (the first radar volume scan when clouds are detected by radar) were compared to determine the early difference between thunderstorms and non-thunderstorms on the basis of this dataset (Zhao et al., 2022). The greater echo intensity occurred in non-thunderstorms below the $-10°C$ isotherm height, and the cause for this feature and effect on subsequent cloud development were simply discussed by integrating comprehensive observations (e.g., the ERA-Interim reanalysis data, surface aerosol concentration, and graupel and rainwater contents derived from radar observations).

The error in the graupel content estimated in Zhao et al. (2022) is uncertain, and the efficiency of the microphysical process (i.e., riming) associated with graupel is unknown; this represents a gap in understanding regarding the role of graupel in the first lightning flash occurrence based on field observations. Naturally, we aimed to identify a method to quantify differences in graupel magnitude and riming efficiency in this study to minimize the error as much as possible. The radar sample volume, which corresponds to graupel identification, was used to indicate the graupel magnitude instead of the derived graupel content, as in Carey and Rutledge (2000) and Zhao et al. (2022). The variety of $Z_{DR}$ shapes was used to determine the riming efficiency. Thus, the goal and method of this study were substantially different from those of the two previous studies noted above, although they are based on the same dataset."

**Lines 278−284** in mms:

"In addition, the average 1-hourly surface concentration observations of particulate matter (PM$_{2.5/10}$) were provided by three ground sites (Figure 1, white diamonds) within the analysed area. The PM$_{2.5/10}$ concentration data suggest that the environment prior to these isolated thunderstorms or non-thunderstorms was clean and that the difference in the environmental aerosol concentration between thunderstorms and non-thunderstorms may be small (the mean values of PM$_{2.5/10}$ concentrations prior to thunderstorms and non-thunderstorms were 22.9/42 µg m$^{-3}$ and 20.5/38.8 µg m$^{-3}$, respectively)."

Zhao, C., Zheng, D., Zhang, Y. J., Liu, X., Zhang, Y., Yao, W., Zhang, W.: Turbulence Characteristics before the Occurrence of the First Flash in Thunderstorms and Non-Thunderstorms, *Geophysical Research Letters*, 48, e2021GL094821, 2021a.

Zhao, C., Zhang, Y. J., Zheng, D., Liu, X., Zhang, Y., Fan, X., Yao, W., Zhang, W.: Using polarimetric radar observations to characterize first echoes of thunderstorms and nonthunderstorms: A comparative study, *Journal of Geophysical Research: Atmospheres*, 127, e2022JD036671, 2022.

*- Abstract: I think authors should define $Z_{DR}$.*

**Reply:** Corrected. The $Z_{DR}$ is defined. Please see mms (Line 39).

**Line 39** in mms:

"…with a mean differential reflectivity ($Z_{DR}$) value of 0.3 dB…"

*- Line 61: Mention in-cloud corona discharges when discussing the different types of lightning activity.*

**Reply:** To avoid confusion, this sentence has been revised. Please see mms (Lines 67−69).

**Lines 67−69** in mms:

"Moreover, natural lightning flashes are generally defined as intracloud lightning and cloud-to-ground lightning (Uman and Krider, 1989)."

*- Lins 69-74: Aerosols play an important role in cloud electrification. Please mention.*

**Reply:** The draft has been revised as suggested. Please see mms (Lines 75−81).

**Lines 75−81** in mms:

"…noninductive charging (NIC) of two ice particles of different sizes during rebounding collisions in the presence of supercooled droplets, with the smaller ice particle being the ice crystal and the larger ice particle being the graupel; aerosol provides the cloud condensation nuclei and ice nuclei for hydrometeor formation, thus playing an important role in cloud electrification (Takahashi, 1978; Latham, 1981; Saunders et al., 1991; MacGorman and Rust, 1998; Carey and Rutledge, 2000; Rosenfeld et al., 2008; Zhang et al., 2009; Takahashi et al., 2017, 2019; Qie et al., 2021; Lyu et al., 2023)."

*- Line 122: Plotting the analysed area could be helpful for the readers.*

**Reply:** We have added the related plot as suggested. Figure 1 shows the analysed area and the locations of the detection systems.

[Figure]

**Figure 1. The locations of the detection systems and the analysed area.** The orange star indicates the Guangzhou S-band polarimetric radar (GZ radar); the orange circles represent distances from the GZ radar site of 25 and 100 km. The black dots indicate the 10 sensors of the Low-Frequency E-field Detection array (LFEDA); the black circle indicates the distance from the centre of the LFEDA network to 70 km. The blue triangles indicate the 16 sensors of the Earth Networks Lightning Location System (ENLLS), and the orange triangles indicate the 27 sensors of the Guangdong Lightning Location System (GDLLS). The white diamonds indicate the three ground sites of aerosol concentration measurements. The orange diamond indicates the Qingyuan meteorological observatory. The analysed area is restricted to the regions of overlapping coverage between the GZ radar radius of 25−100 km and the LFEDA station network centre radius of 70 km.

*- Line 142: Please provide the coverage of LFEDA.*

**Reply:** Yes, we have provided the coverage of LFEDA as suggested. Figure 1 shows the analysed area and the locations of the detection systems.

*- Line 153: This has already been said before.*

**Reply:** Corrected.

*- Line 190: 98% of total FLF are IC. Could you compare this percentage with other studies? I do not say that the authors have to do this, but this could be interesting.*

**Reply:** Thank you for the suggestion. We agree that the comparison could be interesting.

We have added the related content of the lightning observations (Figure 2) and compared the results with those of a previous study (i.e., Mattos et al., 2017). Please see mms (Lines 258−277).

The results of the first IC and/or CG flashes from the three lightning location systems are shown in Figure 2a. The majority of the first flashes are IC flashes (56/57, ~98%), and only one is a CG flash (1/57). Additionally, the majority of the first flashes (~91%) are determined by LFEDA because of the superior detection efficiency and accuracy of LFEDA for lightning flashes within the analysed area.

The elapsed time between the first radar volume scan and the first IC or CG flash (indicating by the first IC or CG return stroke) is shown in Figure 2b. The results show that the average elapsed time between the first radar volume scan and the first IC flash was approximately 19 minutes, and the first CG flash was approximately 32 minutes (Figure 2b). A recent study (Mattos et al., 2017) also revealed that in ~98% of thunderstorms, an IC flash preceded the first CG flash, and the IC flashes occurred approximately 29 minutes after the first radar echo (any reflectivity value (any value above the local noise floor of the radar) at any height), CG flashes were most frequently delayed by approximately 36 minutes. The definition of the first radar echo may be the possible reason that the first flashes occurring after the first radar echo in Mattos et al. (2017) occurred later than those in our study.

[Figure]

[Figure]

**Figure 2. Lightning observations.** Elapsed time between the first radar volume scan and (a) the first flashes of three lightning location systems, LFEDA (red line), ENLLS (blue line), and GDLLS (black line), where the grey circles indicate the first IC flashes, the grey diamonds indicate the first CG flashes, and (b) the elapsed time between the first radar volume scan and the first flashes of thunderstorms, the first IC flashes (black columns), and the first CG flashes (red columns).

*- Results: I miss an analysis of the significance of the differences between thunderstorms and non-thunderstorms cells, which was for example provided in Zhao et al. (2022, GRL).*

**Reply:** We have added an analysis of the significance of the differences between thunderstorms and non-thunderstorms as suggested. Please see in mms (Lines 301−307; 348−365; 383−389; 481−484; 520−521; 556−562; 583−588).

*- Figure 2: Why is the graupel volume larger in non-thunderstorm cells during the first stage? Could you provide an explanation?*

**Reply:** We discovered that the polarimetric radar parameters of the first radar echoes clearly differed between thunderstorms and non-thunderstorms; specifically, the echo intensity was greater in non-thunderstorms below the $-10°C$ isotherm height. Thus, a greater graupel volume in non-thunderstorm cells during the first stage is possible. However, how this phenomenon occurs is uncertain. We speculate that more warm precipitation growth in non-thunderstorms due to cyclic drop growth resulting from coalescence under weaker updrafts may promote greater drops formation (Kumjian et al., 2014; Mather et al., 1986; Stough et al., 2021). These larger drops are lifted above the 0°C isothermal height and freeze to graupel-sized particles via a coalescence-freezing mechanism (e.g., Bringi et al., 1997; Carey and Rutledge, 2000).

A related discussion has been added to the draft. Please see mms (Lines 357−362).

**Lines 357−362** in mms:

"We proposed a mechanism for explaining the larger graupel volume in non-thunderstorms during the first stage of cloud development: more warm precipitation growth in non-thunderstorms due to cyclic drop growth resulting from coalescence under weaker updrafts may promote greater drop formation (Kumjian et al., 2014; Mather et al., 1986; Stough et al., 2021). These larger drops are lifted above the 0°C isothermal height and freeze to graupel-sized particles via a coalescence-freezing mechanism (e.g., Bringi et al., 1997; Carey and Rutledge, 2000)."

Bringi, V. N., Knupp, K., Detwiler, A., Liu, L., Caylor, I. J., and Black, R. A.: Evolution of a Florida Thunderstorm during the Convection and Precipitation/Electrification Experiment: The Case of 9 August 1991, *Monthly Weather Review*, 125, 2131–2160, doi: https://doi.org/10.1175/1520-0493(1997)125<2131:EOAFTD>2.0.CO;2,1997.

Carey, L. D., and Rutledge, S. A.: The Relationship between precipitation and lightning in tropical island convection: A C-Band polarimetric radar study, *Monthly Weather Review*, 128, 2687–2710, https://doi.org/10.1175/1520-0493(2000)128<2687:TRBPAL>2.0.CO;2, 2000.

Kumjian, M. R., Khain, A. P., Benmoshe, N., Ilotoviz, E., Ryzhkov, A. V., and Phillips, V. T. J.: The anatomy and physics of ZDR columns: Investigating a polarimetric radar signature with a spectral bin microphysical model, *Journal of Applied Meteorology and Climatology*, 53, 1820–1843, https://doi.org/10.1175/JAMC-D-13-0354.1, 2014.

Mather, G. K., Morrison, B. J., and Morgan, G. M.: A Preliminary Assessment of the Importance of Coalescence in Convective Clouds of the Eastern Transvaal, *Journal of Applied Meteorology and Climatology*, 25, 1780–1784, https://doi.org/10.1175/1520-0450(1986)025<1780:APAOTI>2.0.CO;2, 1986.

Stough, S. M., Carey, L. D., Schultz, C. J., and Cecil, D. J.: Examining conditions supporting the development of anomalous charge structures in supercell thunderstorms in the Southeastern United States, *Journal of Geophysical Research: Atmospheres*, 126, e2021JD034582, https://doi.org/10.1029/2021JD034582, 2021.

*- Line 259: "...in this volume These characteristics..." -> "...in this volume. These characteristics..."*

**Reply:** Corrected.

---

## Author Comment (AC2)

**Responses to Reviewers' Comments**

We sincerely appreciate the time and effort devoted by the anonymous reviewers and editor. We thank the reviewers for these constructive and professional comments. And Our point-to-point responses can be found below. The reviewer comments/suggestions are in *italic* font, and our responses are underlined and in blue. The file name "Manuscript with marked changes" is abbreviated as "mms".

**Referee #2 Evaluations:**

**General comments**

*The objective of this paper is to «better understand the ice microphysics associated with graupel within thunderstorms». To this aim, 57 isolated thunderstorms and 39 isolated non-thunderstorms over South China in the warm season (2016 and 2017) are studied using the Guangzhou S-pol radar with a hydrometeor classification algorithm, and 3 independant lightning location systems. The investigated storms are divided into 3 stages: first radar volume scanning ($Z_H > 5$ dBZ), intermediate radar volume scanning (between the first and third stages), radar volume scanning where the first lightning flash is detected (for thunderstorms) or where the most intense radar echo is detected (for non-thunderstorms).*

*This paper addresses the microphysical characteristics of the storm at the occurrence of the first lightning flash which is a relevant scientific question. However, technical notes should report «new developments, significant advances, or novel aspects of experimental and theoretical methods and techniques that are relevant for scientific investigations within the scope of the journal». In the submitted manuscript, I was not able to find which development, method or technique is new. Same data and tools as in Zhao et al. (2021) and Zhao et al. (2022) are used. The authors emphasize that their study is based on a large sample statistics of thunderstorms and non-thunderstorms (see their Table 1 where Mattos et al. (2016) is not referenced), and that they based their study on graupel particles inferred by a hydrometeor classification method (also done in previous papers by Zhao et al. among others). This point should be clarified. Also, I wonder why the authors did not use the 3D mapping of lightning flashes from LFEDA to investigate further the relationship between ice microphysics and the first lightning flash (triggering altitude, altitude of the charged regions...).*

**Reply:** We sincerely appreciate your time in reviewing our manuscript. Your professional comments and suggestions are beneficial for us in improving this study. We have added related content to clarify the novelty and motivation of this study. The

observations of 3D lightning mapping from LFEDA are shown in the revised manuscript. In addition, the observational characteristics associated with the source initiation and channel of the first lightning flash are displayed in the revised manuscript. Please see mms (Lines 137−138; 150−175; 258−277; 465−499). The information about the reference (Mattos et al. 2016) is displayed in your comment, fourth from the end. Related discussion in the revised manuscript. Please see mms (Lines 583−588).

This study is substantially different from the two previous studies (i.e., Zhao et al., 2021a, 2022) noted above, although they use the same dataset.

In Zhao et al. (2021a), we first presented the dataset to the public, which included observations of 57 (39) isolated thunderstorms (non-thunderstorms) during 2016/2017 over South China from the S-band polarimetric radar and three independent lightning location systems. The objective of this study was to investigate the turbulence characteristics of thunderstorms before the first flash in comparison to those of non-thunderstorms. We utilized this dataset and derived the eddy dissipation rate from the Doppler velocity to evaluate the role of turbulence characteristics in producing the first lightning flash in the cloud. The results indicated that the eddy dissipation rate of non-thunderstorms was clearly lower than that of thunderstorms.

At the peer review stage of Zhao et al. (2021a), an anonymous reviewer noted the turbulence difference in the first radar volume scan between thunderstorms and non-thunderstorms (i.e., a stronger eddy dissipation rate in non-thunderstorms). However, we also propose the following question: what was the difference between thunderstorms and non-thunderstorms in the first radar volume scan, and did it affect cloud development?

Thus, we utilized this dataset to evaluate the polarimetric radar parameters of the first radar echoes (the first radar volume scan when clouds occurred and were detected by radar) in Zhao et al. (2022). We discovered that the polarimetric radar parameters of the first radar echoes clearly differed between thunderstorms and non-thunderstorms; specifically, a greater echo intensity was present in non-thunderstorms below the − 10°C isotherm height. In addition, the ERA-Interim reanalysis data and surface aerosol concentration observations were used to determine the reason. Finally, the graupel and rainwater contents (the value of the 90% quantile at different altitudes during different development stages of storms) were compared, and the results suggested that the difference in the first radar echoes between thunderstorms and non-thunderstorms may play an important role in subsequent cloud development.

In Zhao et al. (2022), the difference in polarimetric radar parameters in the first radar echoes between thunderstorms and non-thunderstorms was determined. In addition, the graupel content was shown during cloud development to suggest convection invigoration according to latent heat release.

However, the error in graupel content estimation is uncertain, and the efficiency of the microphysical process (i.e., riming) associated with graupel is unknown. Naturally, we want to seek a method to quantify differences in graupel magnitude and riming efficiency, while minimizing the error as much as possible.

Therefore, we accomplish this goal by comparing the ice microphysics associated with graupel between isolated thunderstorms and non-thunderstorms during the warm season over southern China and quantifying differences in graupel magnitude and shape (implying riming efficiency) in radar parameters. The radar sample volume, which corresponds to graupel identification, is used to indicate the graupel magnitude instead of the derived graupel content, as in Carey and Rutledge (2000) and Zhao et al. (2022). The variety of $Z_{DR}$ shapes is used to determine the riming efficiency. In addition, the coalescence-freezing mechanism, which is a generally accepted mechanism for graupel formation in warm-based clouds, is explored for the production of the first lightning flash. The results (i.e., the variety of $Z_{DR}$ shapes) could be compared with those in cold-based clouds (Li et al., 2018).

Moreover, in the revised manuscript, the observational characteristics of the first lightning flashes are shown via 3D lightning mapping from LFEDA. The possible microphysics associated with the source initiation and channel of the first lightning flashes are discussed.

Thus, we believe that the goal and method in this manuscript are substantially different from those in the two papers 
[revised manuscript text omitted]

*The article requires thorough proofreading. Abbreviations (e.g. $Z_{DR}$, $Z_H$) should be defined.*

**Reply:** Corrected.

*Most of the time, the number and quality of references are appropriate. But some references are missing for some statements. For exemple:*

- *lines 83-87: references are missing about coalescence-freezing rime as the main pathway for graupel formation in warm-base clouds*

- lines 105-109: there are 2 different statements: studies about relationships between ice microphysics and lightning activity, and methods to predict the first lightning flash occurrence based on the riming electrification mechanism. Please separate the references associated with the 2 statements. The important work of Latham et al. (2007) should be cited.

- Concerning the non-inductive mechanism (lines 76-80), modelling studies also support this mechanism as the main contributor in charge separation conducive to lightning flash triggering in timescales relevant to storm duration (e.g. Helsdon et al., 2001; Mansell et al., 2005; Barthe and Pinty, 2007).

- The reference Boggs et al. (2002) is not relevant for tropospheric lightning flashes; it rather deals with upward electrical discharges from thunderstorms.

**Reply:** Thank you for your careful reading and sound advice. The draft has been revised as suggested. Please see mms (Lines 93−95; 113−123; 87−89).

**Specific comments**

line 85: what is «supercooled temperature»?

**Reply:** We have corrected this description. Please see mms (Lines 95−96).

**Lines 95−96** in mms:

"…followed by lofting of the rain drop in the updraft to subfreezing temperatures…"

line 109-110: add a reference for this statement

**Reply:** Yes, the references have been added. Please see mms (Lines 119−123).

**Lines 119−123** in mms:

"…and provided methods for predicting the first lightning flash occurrence based on the riming electrification mechanism; specifically, graupel-related reflectivity at −10°C or colder is a commonly supported leading reflectivity parameter for forecasting the first lightning flash (e.g., Laksen and Stansbury, 1974; Marshall and Radhakant, 1978; Vincent et al., 2003; Woodard et al., 2012; Hayashi et al., 2021)."

lines 171-173: a map showing the analysis area would be appreciated, even if it is already shown in Zhao et al. (2021, 2022)

**Reply:** Thank you for the suggestion. Figure 1 has been added, showing the analysed area and the locations of the detection systems.

[Figure]

**Figure 1. The locations of the detection systems and the analysed area.** The orange star indicates the Guangzhou S-band polarimetric radar (GZ radar); the orange circles represent distances from the GZ radar site of 25 and 100 km. The black dots indicate the 10 sensors of the Low-Frequency E-field Detection array (LFEDA); the black circle indicates the distance from the centre of the LFEDA network to 70 km. The blue triangles indicate the 16 sensors of the Earth Networks Lightning Location System (ENLLS), and the orange triangles indicate the 27 sensors of the Guangdong Lightning Location System (GDLLS). The white diamonds indicate the three ground sites of aerosol concentration measurements. The orange diamond indicates the Qingyuan meteorological observatory. The analysed area is restricted to the regions of overlapping coverage between the GZ radar radius of 25−100 km and the LFEDA station network centre radius of 70 km.

*line 204: the average duration between the 1st and the 3rd stage is 19 and 24 min, respectively, for thunderstorms and non-thunderstorms. However, the definition of the 3rd stage is different for these two types of storms. It would be interesting to compare the duration between the 1st stage and the occurrence of the maximum radar echo.*

**Reply:** Yes. It would be interesting to compare the duration between the 1st stage and the occurrence of the maximum radar echo. However, in this study, we want to better understand the ice microphysics associated with graupel and how to produce the first

lightning flash. Thus, we selected these three stages to investigate the differences in ice microphysics associated with graupel between isolated thunderstorms and non-thunderstorms. If we continue to investigate the overall life cycle evolution of thunderstorms and non-thunderstorms, a comparison of the duration between the 1st stage and the occurrence of the maximum radar echo should be performed. We will continue to study the microphysics that produce lightning and affect lightning activity in future work.

*line 211: there is no «grey triangles» in Figure 1a*

**Reply:** Corrected. Please see mms (Line 288).

**Line 288** in mms:

"The scatters and triangles with error bars in Figure 3a…"

*line 216: the -30°C isotherm is not shown on Figure 1*

**Reply:** Corrected. "Figure 1" in the raw manuscript has been replaced by "Figure 3" in the revised manuscript.

**Lines 319−330** in mms:

[Figure]

**Figure 3. Characteristics of radar echoes with cloud development.** (a) Echo-top heights of 0 dBZ and echo-base heights of 0 dBZ for 57 thunderstorm and 39 non-thunderstorm cells from the first stage to the third stage of cloud development are indicated by scatter points and triangles, respectively, with error bars. Error bars are computed as 95% confidence intervals. Box plots for the 57 thunderstorms (orange) and 39 non-thunderstorms (blue) for echo depths; all units are in km. The dashed grey lines indicate the −38°C and −30°C isotherm heights. (b) The mean (maximum) value of the $Z_H$ in a thunderstorm or a non-thunderstorm during every stage is shown in notched box plots (non-notched box plots), with all units in the dBZ. The median values in the box plots are shown as black horizontal continuous lines. The temperature data were obtained from the sounding data of the Qingyuan meteorological observatory."

line 221: «deep convective clouds, indicated by thunderstorms, were formed when first lightning flashes occurred»: does it mean that those cloud are not considered as deep convective if there's no lightning activity?

**Reply:** Lightning is the product of a severe storm. Scientists often equate storm intensity with lightning flashes (Zipser et al., 2006). Some studies also indicate that

deep convective clouds form with lightning flashes in their conceptual model (e.g., Fan et al., 2018). However, this is an interesting issue, as described in Zipser et al. (2006): defining intensity is not as easy as it may seem; a search for "intense convection" will find over 1,000 American Meteorological Society (AMS) papers that use this adjective for convective storms. In addition to the lighting flash rate, many papers implicitly equate the intensity with the updraft magnitude. If the subject is subjected to severe weather analysis or forecasting, the National Weather Service definition is often used or implied, requiring wind gusts > 25 m s$^{-1}$, hail > 1.9 cm in diameter, or a tornado.

Thus, in this study, we indicate that the first occurrence of lightning in isolated thunderstorms could mark the formation of deep convective clouds on the basis of a comparison with isolated non-thunderstorms. To avoid the confusion, we have revised the description of this sentence. Please see mms (Lines 301−307).

**Lines 301−307** in mms:

"When the first lightning flashes occurred, approximately 84% of the thunderstorms and only 23% of the non-thunderstorms achieved an echo depth of 10 km. Lightning is the product of the severe storms, and scientists often equate storm intensity with lightning flashes (e.g., Zipser et al., 2006; Fan et al., 2018), but defining convective intensity is not as easy as it may seem (Zipser et al., 2006); this could provide supplementary quantitative evidence for assisting scientists in equating storm intensity with lightning flashes and determining the cloud depth corresponding to the first lightning flash occurrence."

Fan, J. W., Rosenfeld, D., Zhang, Y., Giangrande, S. E., Li, Z., Machado, L. A. T., Martin, S. T., Yang, Y., Wang, J., Artaxo, P., Barbosa, H. M. J., Braga, R. C., Comstock, J. M., Feng, Z., Gao, W., Gomes, H. B., Mei, F., Pöhlker, C., Pöhlker, M. L., Pöschl, U., and de Souza, R. A. F.: Substantial convection and precipitation enhancements by ultrafine aerosol particles, *Science*, 359, 411–418, DOI: 10.1126/science.aan8461, 2018.

Zipser, E. J., Cecil, D. J., Liu, C., Nesbitt, S. W. and Yorty, D. P.: WHERE ARE THE MOST INTENSE THUNDERSTORMS ON EARTH? *Bulletin of the American Meteorological Society*, 87, 1057–1072, https://doi.org/10.1175/BAMS-87-8-1057, 2006.

*Line 225: «values… are slight». Give values to justify this statement.*

**Reply:** Corrected. Please see mms (Lines 308−311).

**Lines 308−311** in mms:

"Figure 3b shows that the differences in the mean (maximum) values of the $Z_H$ between the thunderstorm and non-thunderstorm periods during each stage are slight; specifically, the median differences in the mean values are −2, 2, and 3 dBZ, respectively. The median differences in the maximum values are −4, 5, and 5 dBZ, respectively."

*line 239: the 0°C isotherm is not plotted in Figure 1*

**Reply:** Corrected. Please see mms (Line 326) and Figure 3.

*lines 247-250: this sentence and the causal link with the previous one are not clear*

**Reply:** Corrected. Please see mms (Lines 332−337).

**Lines 332−337** in mms:

"Graupel is a vital precipitation particle for the riming electrification mechanism, and its radar signature is not obscured by small ice particles. Thus, to investigate the microphysical characteristics related to the first lightning flash occurrence during storms, we obtained inferred "graupel", which was derived from the fuzzy-logic method based on the GZ radar (Park et al., 2009; Kumjian, 2013; Zhao et al., 2021b, 2022)."

*lines 251-254: not clear how the graupel volume is computed: please clarify. What is a height layer?*

**Reply:** Corrected. Please see mms (Lines 338−342).

**Lines 338−342** in mms:

"Each bar in Figure 4 indicates the mean value of the graupel volume in a height layer (the definition of the height layer is a vertical resolution of 500 m over 0.5 to 20 km above the mean sea level, 40 height layers in total) for 57 thunderstorms or 39 non-thunderstorms during each stage of cloud development. Specifically, the volume is computed by accumulating the radar sample grids; each radar sample grid is 0.03125 $km^3$, 0.25 km×0.25 km×0.5 km."

*Lines 254-255: «graupel is rare in thunderstorms or non-thunderstorms during the first stage of cloud development». As stated in Lang and Rutledge (2011), «the existence of 30-dBZ echo above the freezing altitude is a "necessary" condition (in ~90% of cases)*

*for lightning occurrence». This value is well above the 5 dBZ threshold used in this study to detect the 1st stage of the storm and can explain why graupel is rare in stage 1. From a modeling study of an isolated thunderstorm, Barthe and Pinty (2007) showed a delay ~ 20 min between the first occurrence of graupel and the first lightning flash (see their Figure 1). In this case study, this delay was attributed to the time for graupel and vapor-grown ice to locally gain charge through the non-inductive mechanism, and to the sedimentation of the different particles leading to macroscopic charge separation.*

**Reply:** Thank you for your suggestion to explain why rare graupel is present in thunderstorms or non-thunderstorms during the first stage of cloud development. We have added a supplementary explanation to the revised manuscript. Please see mms (Lines 346−356).

**Lines 346−356** in mms:

"Graupel is rare in thunderstorms or non-thunderstorms during the first stage of cloud development (e.g., Dye et al., 1986; Mattos et al., 2017), and only 5% (13%) of thunderstorms (non-thunderstorms) show graupel signals (Figure 4). This finding is consistent with the results of Lang and Rutledge (2011), who indicated that the existence of a 30 dBZ echo above the freezing altitude is a necessary condition (in ~90% of cases) for lightning occurrence. This value is well above the 5 dBZ threshold used in this study to detect the first stage of a storm and can explain why graupel is rare in this stage. Moreover, in a modelling study of an isolated thunderstorm, Barthe and Pinty (2007) reported a delay of ~20 minutes between the first occurrence of graupel and the first lightning flash. In this case study, this delay was attributed to the time for graupel and vapour-grown ice to locally gain charge through the NIC mechanism and to the sedimentation of the different particles leading to macroscopic charge separation."

*line 256: how to explain the larger graupel volume in non-thunderstorms during stage 1?*

**Reply:** Yes, this is an interesting issue. Zhao et al. (2022) reported that the polarimetric radar parameters of the first radar echoes clearly differed between thunderstorms and non-thunderstorms; specifically, the echo intensity was greater in non-thunderstorms below the $-10°C$ isotherm height. Thus, a greater graupel volume in non-thunderstorm cells during the first stage is possible. However, how this phenomenon occurs is uncertain. We speculate that more warm precipitation growth in non-thunderstorms due to cyclic drop growth resulting from coalescence under weaker

updrafts may promote greater drop formation (Kumjian et al., 2014; Mather et al., 1986; Stough et al., 2021). These larger drops are lifted above the 0°C isothermal height and freeze to graupel-sized particles via a coalescence-freezing mechanism (e.g., Bringi et al., 1997; Carey and Rutledge, 2000).

A related discussion has been added to the draft. Please see mms (Lines 357−362).

**Lines 357−362** in mms:

"We proposed a mechanism for explaining the larger graupel volume in non-thunderstorms during the first stage of cloud development: more warm precipitation growth in non-thunderstorms due to cyclic drop growth resulting from coalescence under weaker updrafts may promote greater drop formation (Kumjian et al., 2014; Mather et al., 1986; Stough et al., 2021). These larger drops are lifted above the 0°C isothermal height and freeze to graupel-sized particles via a coalescence-freezing mechanism (e.g., Bringi et al., 1997; Carey and Rutledge, 2000)."

Bringi, V. N., Knupp, K., Detwiler, A., Liu, L., Caylor, I. J., and Black, R. A.: Evolution of a Florida Thunderstorm during the Convection and Precipitation/Electrification Experiment: The Case of 9 August 1991, *Monthly Weather Review*, 125, 2131–2160, doi: https://doi.org/10.1175/1520-0493(1997)125<2131:EOAFTD>2.0.CO;2,1997.

Carey, L. D., and Rutledge, S. A.: The Relationship between precipitation and lightning in tropical island convection: A C-Band polarimetric radar study, *Monthly Weather Review*, 128, 2687–2710, https://doi.org/10.1175/1520-0493(2000)128<2687:TRBPAL>2.0.CO;2, 2000.

Kumjian, M. R., Khain, A. P., Benmoshe, N., Ilotoviz, E., Ryzhkov, A. V., and Phillips, V. T. J.: The anatomy and physics of ZDR columns: Investigating a polarimetric radar signature with a spectral bin microphysical model, *Journal of Applied Meteorology and Climatology*, 53, 1820–1843, https://doi.org/10.1175/JAMC-D-13-0354.1, 2014.

Mather, G. K., Morrison, B. J., and Morgan, G. M.: A Preliminary Assessment of the Importance of Coalescence in Convective Clouds of the Eastern Transvaal, *Journal of Applied Meteorology and Climatology*, 25, 1780–1784, https://doi.org/10.1175/1520-0450(1986)025<1780:APAOTI>2.0.CO;2, 1986.

Stough, S. M., Carey, L. D., Schultz, C. J., and Cecil, D. J.: Examining conditions supporting the development of anomalous charge structures in supercell thunderstorms in the Southeastern United States, *Journal of Geophysical Research: Atmospheres*, 126, e2021JD034582, https://doi.org/10.1029/2021JD034582, 2021.

line 257: «is reached» → reaches

**Reply:** Corrected.

lines 263-264: «the black dots … in km$^3$»: not clear, please rephrase.

**Reply:** Corrected. Please see mms (Lines 369−372).

**Lines 369−372** in mms:

"Each grey dot indicates the total graupel volume on a height layer (the definition of the height layer is a vertical resolution of 500 m over 0.5 to 20 km above the mean sea level, 40 height layers in total) of a thunderstorm; the black dots indicate non-thunderstorms (units in km$^3$)."

Lines 271-278: the graupel volume is also found in upper layers in thunderstorms compared to non-thunderstorms. For non-thunderstorms, graupel volume is only found below 4 km above the melting layer, while it reaches altitudes well above the -38°C isotherm in thunderstorms. Mattos et al. (2016) observed different radar signatures in the glaciated part of thunderstorms and non-thunderstorms during the CHUVA field campaign. It could have been discussed.

**Reply:** Yes, the comparison between this study and similar studies (e.g., Mattos et al., 2016) is interesting. However, Mattos et al. (2016) constructed a radar and lightning colocated dataset on the basis of a three-dimensional Cartesian box. This three-dimensional Cartesian box had a grid cell spacing of 1 km × 1 km in the horizontal direction and 15 vertical levels of 1 km in the vertical direction (hereafter called the grid box). If radar signatures are in the grid box but there are no VHF sources, the grid box is regarded as a "NOVHF" event. The definition of non-thunderstorm events in this study is different from the definition of "NOVHF" events in Mattos et al. (2016). In addition, we speculate that the radar signatures in the glaciated part of the NOVHF event during the CHUVA field campaign may have resulted from cloud anvil.

We added a related discussion to the revised manuscript. Please see mms (Lines 583−588).

**Lines 583−588** in mms:

"In addition, unlike previous similar studies (e.g., Mattos et al., 2016, 2017), we studied the microphysical differences between isolated thunderstorms and non-thunderstorms during the warm season over southern China on the basis of polarimetric radar and lightning mapping array instead of studying the evolution variation within the same thunderstorm (Mattos et al., 2017) or studying the differences between storm vertical profiles in three-dimensional Cartesian boxes with lightning and without lightning (Mattos et al., 2016)."

*Lines 289-301: add relevant references in this paragraph.*

**Reply:** Yes, relevant references have been added to this paragraph. Please see mms (Lines 403−413).

*Line 304: what are «the average intensities of the $Z_H$ and $Z_{DR}$»? How are they computed?*

**Reply:** Corrected. Please see mms (Lines 419−424).

**Lines 419−424** in mms:

"The mean values of $Z_H$ and $Z_{DR}$ corresponding to graupel particles (the radar sample grids are identified as graupel) above the ~−3°C isotherm height (avoiding melting effects) in thunderstorms and non-thunderstorms during each stage of cloud development are displayed in Figure 5. Each orange dot indicates the mean values of $Z_H$ and $Z_{DR}$ corresponding to graupel above the ~−3°C isotherm height in a thunderstorm; each blue dot indicates that in a non-thunderstorm."

*Line 327: the differences in graupel formation for winter snowstorms and warm-season thunderstorms should be make clear.*

**Reply:** Thank you for the suggestion. The draft has been revised as suggested. Please see mms (Lines 446−452).

**Lines 446−452** in mms:

"The results from Li et al. (2018) are limited to only winter snowstorms; the mechanism for producing graupel in winter snowstorms is initiated via the aggregation of ice crystals into snow aggregates, followed by riming of the snow aggregate into graupel and possibly even small hail as the rime density increases (Heymsfield, 1982; Li et al., 2018). This process is different from the coalescence-freezing mechanism in warmseason thunderstorms, but the final shape of the graupel particles when first lightning flashes occurred in this study approached the shape of moderately or heavily rimed ice particles in Li et al. (2018)."

---

## Referee Report (RR1)

**Review of « Technical note : On the ice microphysics of isolated thunderstorms and non-thunderstorms in southern China : A radar polarimetric perspective » by Zhao et al.**

The authors have provided pertinent responses to my comments. In particular, the innovative aspect of this study compared with previous ones is now discussed and justified. I also appreciated the inclusion of a discussion of the delay between the detection of the first radar volume and the first flash as seen by the 3 lightning detection networks, and about the characteristics of the lightning sources for the first lightning flash.

Below are a few minor points to be corrected. A detailed re-reading of the manuscript is still necessary.

**Specific comments**

lines 67-68 : natural lightning flashes are not defined as intracloud and cloud-to-ground ; they can be categorized as intranuage or cloud-to-ground.

Line 73 : aadition → addition

lines 77-79 : remove « the aerosol… in cloud electrification » ; this sentence makes no sense here.

Lines 82-86 : make two sentences

line 113 : do you mean « polarimetric radar is the best observation system for tracking … » ?

lines 152-156 : make two sentences

line 168 : remove « regarding »

line 188-189 : the black circle indicates a distance of 70 km from the centre of the LFEDA network

line 265-269 : this sentence is not clear. Please rephrase this sentence.

Figure 2 : most of the time, the time of the first flash occurrence in the LFEDA and ENLLS datasets are the same. However, for some thunderstorms, the delay between the first flash detection with these two systems can reach 20 min. How can you explain this fact ?

Lines 481-482 : please explain this statement.

Lines 481-484 : make two sentences

Figure 6a : « Altitude (km above AGL) » → « Altitude (km AGL) »

---

## Author Response (AR2)

**Responses to Reviewers' Comments**

We sincerely appreciate the time and effort devoted by the anonymous reviewers and editor. We thank the reviewers for these constructive and professional comments again. And Our point-to-point responses can be found below. The reviewer comments/suggestions are in *italic* font, and our responses are underlined and in blue. The file name "Manuscript with marked changes" is abbreviated as "mms".

**Referee #2 Evaluations:**

*The authors have provided pertinent responses to my comments. In particular, the innovative aspect of this study compared with previous ones is now discussed and justified. I also appreciated the inclusion of a discussion of the delay between the detection of the first radar volume and the first flash as seen by the 3 lightning detection networks, and about the characteristics of the lightning sources for the first lightning flash.*
*Below are a few minor points to be corrected. A detailed re-reading of the manuscript is still necessary.*

***Specific comments***
*Lines 67-68: natural lightning flashes are not defined as intracloud and cloud-to ground; they can be categorized as intranuage or cloud-to-ground.*
**Reply:** The draft has been revised as suggested. Please see mms (Lines 67−69).

**Lines 67−69** in mms:

"Moreover, natural lightning flashes can be categorized as intracloud lightning and cloud-to-ground lightning (Uman and Krider, 1989)."

*Line 73: aadition → addition*
**Reply:** Corrected.

*Lines 77-79: remove < the aerosol… in cloud electrification >; this sentence makes no sense here.*
**Reply:** Corrected.

*Lines 82-86: make two sentences*
**Reply:** The draft has been revised as suggested. Please see mms (Lines 80−84).

**Lines 80−84** in mms:

"The NIC was proposed on the basis of cold-chamber laboratory experiments (Reynolds et al., 1957; Takahashi, 1978). Subsequently, field observations demonstrated that lightning production is critically linked to ice processes (i.e., graupel signatures) (Dye et al., 1986; Takahashi et al., 1999; Carey and Rutledge, 2000; Basarab et al., 2015; Stolzenburg et al., 2015; Mattos et al., 2016, 2017; Takahashi et al., 2017, 2019; Hayashi et al., 2021; Zhao et al., 2022)."

*Line 113: do you mean < polarimetric radar is the best observation system for tracking …>?*
**Reply:** Yes, this sentence has been corrected.

*Lines 152-156: make two sentences*
**Reply:** The draft has been revised as suggested. Please see mms (Lines 147−151).

**Lines 147−151** in mms:

"In Zhao et al. (2021a), the dataset was first shown to the public. They obtained observations of 57 (39) isolated thunderstorms (non-thunderstorms) that occurred over South China in the warm season (from late May to early September) of 2016 and 2017 from the S-band polarimetric radar and three independent lightning location systems."

*Line 168: remove <regarding>*
**Reply:** The draft has been revised as suggested.

*Lines 188-189: the black circle indicates a distance of 70 km from the centre of the LFEDA network*
**Reply:** The draft has been revised as suggested. Please see mms (Lines 183−184).

**Lines 183−184** in mms:

"…the black circle indicates a distance of 70 km from the centre of the LFEDA network."

*Lines 265-269: this sentence is not clear. Please rephrase this sentence.*
**Reply:** The draft has been revised. Please see mms (Lines 258−262).

**Lines 258−262** in mms:

"A recent study (Mattos et al., 2017) also revealed that in ~98% of thunderstorms, the first IC flash preceded the first CG flash, and the IC flashes occurred approximately 29

minutes after the first radar echo, CG flashes were most frequently delayed by approximately 36 minutes."

*Figure 2: most of the time, the time of the first flash occurrence in the LFEDA and ENLLS datasets are the same. However, for some thunderstorms, the delay between the first flash detection with these two systems can reach 20 min. How can you explain this fact?*

**Reply:** It depends on the detection efficiency of the lightning location system. As described in context, the LFEDA has the superior detection efficiency in this analysis area. Thus, if the LFEDA detects the accurate first flash but the ENLLS missed it, and the ENLLS takes the later lightning flash for the first flash, the delay between the first flash detection with the LFEDA and ENLLS can reach 20 min.

*Lines 481-482: please explain this statement.*

**Reply:** We have rephrased this sentence. The heights of the initiation sources and propagation sources of the first lightning flashes within isolated thunderstorms over Guangzhou are concentrated at an approximate −10°C isotherm height, which provides supplementary evidence that the main negative charge layer is located at −10°C to −20°C isotherm height on Earth, as reported by Krehbiel (1986). Please see mms (Lines 462−466).

**Lines 462−466** in mms:

"The heights of the initiation sources and propagation sources of the first lightning flashes within isolated thunderstorms over Guangzhou are concentrated at an approximate −10°C isotherm height, which provides supplementary evidence that the main negative charge layer is located at −10°C to −20°C isotherm height on Earth, as reported by Krehbiel (1986)."

*Lines 481-484: make two sentences*

**Reply:** The draft has been revised as suggested. Please see mms (Lines 462−469).

**Lines 462−469** in mms:

"The heights of the initiation sources and propagation sources of the first lightning flashes within isolated thunderstorms over Guangzhou are concentrated at an approximate −10°C isotherm height, which provides supplementary evidence that the main negative charge layer is located at −10°C to −20°C isotherm height on Earth, as reported by Krehbiel (1986). The values of $Z_H$ ($Z_{DR}$) corresponding to the initiation

sources and propagation sources of the first lightning flashes suggest that are differences in particle shape and/or size between initiation sources and propagation sources, although the differences are too subtle to quantify in this study."

*Figure 6a: <Altitude (km above AGL)> → <Altitude (km AGL)>*
**Reply:** Corrected.